# SAGE-NAS: Synergizing LLM-Based Semantic Agent with Graph-Based Evaluator for Neural Architecture Search

**Kaiqi Lin** [1]   **Jianping Luo** [1]

## Abstract

While LLM-driven Neural Architecture Search (NAS) leverages exceptional code generation and reasoning, it suffers from a critical "Semantic-Physical Misalignment": LLMs prioritize high-level semantic plausibility but are agnostic to intrinsic physical dynamics such as gradient flow, whereas Zero-Cost Proxies (ZCPs) capture these local sensitivities but lack global semantic planning. To bridge this gap, we propose SAGE-NAS, a closed-loop evolutionary framework that synergizes an LLM-Based Semantic Agent with a Graph-Based Evaluator. Specifically, SAGE-NAS coordinates an LLM-driven Semantic Agent to construct candidate architectures by dynamically scheduling complementary sub-policies that balance exploitation with exploration. Furthermore, the framework integrates a Dual-Modality Graph Evaluator that serves as a rapid performance predictor by fusing ZCP statistics with topological features, and a State-Aware Behavioral Atlas that guides sparsity-driven exploration to escape local optima. Experiments demonstrate that SAGE-NAS achieves state-of-the-art performance across multiple mainstream search spaces and downstream tasks, exhibiting a superior balance between search efficiency, model accuracy, and cross-task generalization capability.

## 1. Introduction

Neural Architecture Search (NAS) has marked a paradigm shift from manual feature engineering to automated exploration, achieving performance surpassing human designs in tasks such as image recognition, object detection, and semantic segmentation (Zoph & Le, 2017; Zela et al., 2018; Chen et al., 2018; Ghiasi et al., 2019). While early approaches based on Reinforcement Learning (RL) and Evolutionary Algorithms (EA) laid the foundation (Zoph et al., 2018; Real et al., 2019), their prohibitive computational costs have steered the community toward more efficient solutions. On one hand, Zero-Cost Proxies (ZCPs) have emerged to rapidly evaluate architectures via statistics such as gradient norms (Abdelfattah et al., 2021), NWOT (Mellor et al., 2021), or SWAP (Peng et al., 2024). However, these ZCPs often suffer from poor rank correlation with final performance due to their localized perspective (Krishnakumar et al., 2022). On the other hand, the rise of Large Language Models (LLMs) has catalyzed Generative NAS (Yu et al., 2023; Nasir et al., 2024), leveraging code generation capabilities for exploration (Chen, 2021). Nevertheless, existing LLM-NAS methods (Chen et al., 2023; Nasir et al., 2024) face a critical "Semantic-Physical Misalignment": LLMs excel in high-level semantic planning but remain agnostic to the intrinsic physical dynamics (e.g., gradient flow and signal propagation) (Tanaka et al., 2020), whereas ZCPs capture these local sensitivities but lack global semantic planning. This cross-modal gap often leads to "hallucinated" architectures—designs that are semantically plausible in code but optimization-hostile in practice.

To bridge this gap, we propose SAGE-NAS (Synergizing LLM-Based Semantic Agent with Graph-Based Evaluator for NAS). Unlike approaches treating LLMs as black-box optimizers or ZCPs as isolated filters, SAGE-NAS establishes a closed-loop framework that aligns semantic reasoning with physical evaluation. Specifically, SAGE-NAS coordinates an LLM-driven Semantic Agent to construct candidate architectures by dynamically scheduling complementary sub-policies that balance exploitation with exploration. To rapidly validate these candidates, we integrate a Dual-Modality Graph Evaluator that fuses ZCP statistics (physical priors) with graph features via cross-attention, explicitly synergizing topological structures with their intrinsic optimization dynamics. Furthermore, to mitigate the mode collapse inherent in LLM generation and escape local optima, we introduce a State-Aware Behavioral Atlas. This module maps architectures into a behavioral space and employs a sparsity-driven mechanism to guide the Semantic Agent toward sparsely explored yet high-potential regions.

[1]Shenzhen Key Laboratory of Media Security, Shenzhen University, Shenzhen, China. Correspondence to: Jianping Luo <ljp@szu.edu.cn>.

The contributions of this work are summarized as follows:

- We propose the SAGE-NAS framework, which resolves the Semantic-Physical Misalignment by synergizing the Semantic Agent with the Dual-Modality Graph Evaluator, ensuring that the generated architectures are both logically sound and practically trainable.
- We design a Dual-Modality Graph Evaluator for rapid performance assessment by fusing topology with physical priors, and a State-Aware Behavioral Atlas to counteract LLM inductive bias via sparsity-driven exploration.
- Extensive experiments demonstrate that SAGE-NAS outperforms existing ZCP and LLM-NAS methods across multiple mainstream search spaces and diverse vision tasks, validating its superior search efficiency, accuracy, and cross-task generalization.

## 2. Related Work

### 2.1. LLM for Neural Architecture Search

The integration of Large Language Models (LLMs) has catalyzed a generative design paradigm in NAS, evolving through three stages. In the early Generative Paradigm, GPT-NAS (Yu et al., 2023) leverages LLM priors for text-based black-box optimization. In the Evolutionary Paradigm, methods like EvoPrompting (Chen et al., 2023) and LLMatic (Nasir et al., 2024) integrate LLMs as evolutionary operators, exploiting code syntax understanding to execute semantically coherent crossover and mutation. The latest Reasoning and Discovery Paradigm emphasizes deep logical interaction. For instance, AutoProxy (Kang et al., 2025) utilizes LLM reasoning to automatically discover optimal Zero-Cost Proxies, while RZ-NAS (Ji et al., 2025) introduces reflection modules to dynamically refine search strategies by parsing proxy feedback. This phase marks a transition of LLMs from generators to thinkers. However, these interactions remain confined to isolated numerical feedback, failing to perceive the geometric landscape of the search space, thereby limiting the LLM to inefficient, local trial-and-error. SAGE-NAS bridges this gap by introducing a Behavioral Atlas to model the geometric search state. Guided by the Behavioral Atlas, the LLM-driven Semantic Agent transforms blind search into strategic navigation by dynamically scheduling complementary sub-policies.

### 2.2. Efficient NAS Evaluation: Proxies and Predictors

To alleviate the training burden in NAS, efficient performance estimation has evolved into two main paradigms: Zero-Cost Proxies (ZCPs) and Model-Based Predictors. ZCPs perform training-free estimation via initialization statistics. Early proxies characterize weight sensitivity or activation patterns (Lee et al., 2019; Wang et al., 2020; Tanaka et al., 2020; Mellor et al., 2021), whereas recent advancements focus on ensemble efficacy, parametric flexibility, and broader evaluation dimensions: AZ-NAS (Lee & Ham, 2024) assembles complementary proxies to enhance ranking correlation, ParZC (Dong et al., 2025a) introduces parametric adaptations for cross-task adaptability, and TRNAS (Yang et al., 2025) extends zero-cost estimation to directly evaluate adversarial robustness. In parallel, GNN-based predictors (Wen et al., 2020) fit architecture-performance mappings on graph structures. Recent approaches such as SemiNAS (Luo et al., 2020) employ semi-supervised learning for robust representations, while GNN-Enhanced Transformers (Xiang et al., 2024) capture global structural dependencies to mitigate over-smoothing. However, both paradigms face inherent limitations: ZCPs, as coarse-grained inductive biases, often suffer from rank inconsistency with final accuracy, while GNN predictors are constrained by cold-start issues and data inefficiency. To resolve this "efficiency-accuracy" dilemma, SAGE-NAS proposes a Dual-Modality Graph Evaluator. By leveraging cross-attention to inject ZCPs as physical priors into topological features, we explicitly synergize structural embeddings with intrinsic optimization dynamics, achieving a robust predictor even under low-data regimes.

## 3. Our Methodology

### 3.1. SAGE-NAS Framework

Conventionally, Neural Architecture Search (NAS) is formulated as a bi-level optimization problem, aiming to identify the optimal architecture that minimizes validation loss based on weights optimized on the training data. However, integrating LLMs introduces a critical "Semantic-Physical Misalignment": LLMs operate on discrete semantic sequences but remain agnostic to the continuous intrinsic physical dynamics (e.g., gradient flow) crucial for trainability, whereas physical proxies lack high-level planning capabilities.

To bridge this cross-modal disparity, SAGE-NAS reformulates the task as a State-Aware Optimization problem governed by the triad $\langle \mathcal{A}_\pi, \mathcal{E}_\phi, \mathcal{M}_{state} \rangle$ (as in Figure 1):

- Semantic Agent $\mathcal{A}_\pi$: Leverages LLMs to execute context-guided architecture evolution (Sec. 3.2).
- Graph-based Evaluator $\mathcal{E}_\phi$: Serves as a Dual-Modality Graph Evaluator that synergizes topological structures with physical priors for rapid and accurate performance estimation (Sec. 3.3).
- Behavioral Atlas $\mathcal{M}_{state}$: Functions as a global state manager to guide sparsity-driven exploration (Sec. 3.4).

SAGE-NAS optimization is a synergistic process involving architecture search and evaluator calibration. We formulate this as a bi-level cooperative optimization problem for

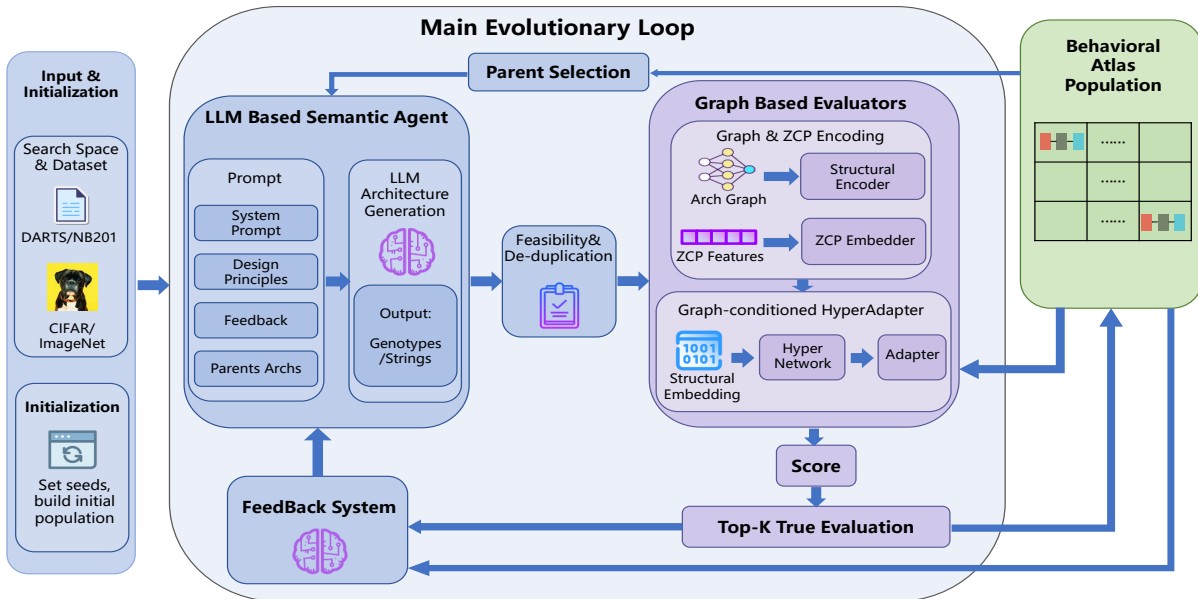

*Figure 1.* The SAGE-NAS Framework. The system consists of three core components: the Semantic Agent, the Graph-Based Evaluator, and the Behavioral Atlas. The cycle involves semantic generation, physics-aware evaluation, and tri-level feedback updates.

architecture $\alpha$ within search space $\mathcal{S}$:

$$\max_{\alpha \in \mathcal{S}} \mathcal{J}_{gt}(\alpha)$$
$$\text{s.t. } \phi^* = \arg\min_{\phi} \mathcal{L}_{rank}\Big(\mathcal{E}_\phi(\mathcal{P}_{hist}), \mathcal{J}_{gt}(\mathcal{P}_{hist})\Big) \quad (1)$$

where the upper-level objective seeks the global optimum $\alpha^*$ that maximizes the ground-truth performance $\mathcal{J}_{gt}$. The lower-level constraint requires evaluator parameters $\phi^*$ to continuously minimize the ranking loss $\mathcal{L}_{rank}$ over the entire evolving history $\mathcal{P}_{hist}$. By explicitly penalizing the discrepancy between predicted and ground-truth rankings across the full dataset, the optimized evaluator $\mathcal{E}_{\phi^*}$ rapidly evaluates generated candidates to provide high-fidelity relative guidance for the upper-level search. Algorithm 1 details this synergistic procedure, which unfolds in four phases:

- Initialize (Lines 1-7): $\mathcal{A}_\pi$ first synthesizes a large heterogeneous pool $\mathcal{P}_{pool}$ via zero-shot generation to broadly cover the search space. We then map these candidates into the Behavioral Atlas $\mathcal{M}_{state}$ by computing their behavioral coordinates. Based on this established spatial distribution, we execute the SelectMaxDispersion procedure (detailed in Appendix B.4) to sample a subset $\mathcal{P}_0$ via Grid-Based Hybrid Sampling. This ensures the evaluator $\mathcal{E}_\phi$ is trained on a geometrically diverse dataset (see ablation in Appendix C.7).
- Context-Guided Semantic Generation (Lines 9-19): Instead of blind mutation, SAGE-NAS executes context-guided batch generation. The system first computes an adaptive scheduling probability $p_{sched}$ based on the atlas coverage rate $\Omega_t$. Subsequently, the process enters an ex-

plicit generation loop for $N$ candidates. In each iteration, a specific sub-policy $\pi_i$ is stochastically sampled based on $p_{sched}$. Crucially, the *Geometric-Semantic Translator* converts the current spatial state (either sparse regions or elite clusters) into actionable feedback $\mathcal{F}_i$. This feedback is dynamically structured by sub-policy $\pi_i$ to construct a candidate-specific prompt $\mathcal{C}_i$, guiding $\mathcal{A}_\pi$ to synthesize a unique architecture $\alpha_{new}$ that precisely executes the intended navigational strategy.
- Dual-Modality Scoring & Selection (Lines 20-22): To avoid expensive training, the system utilizes $\mathcal{E}_\phi$ for rapid estimation. Internally, it extracts topological structures $\mathcal{G}$ and zero-cost proxies $\mathbf{Z}_{phy}$ from the candidate batch $\mathcal{A}_{cand}$ to predict a set of fitness scores $\mathcal{S}_{cand}$. Based on these predictions, the system identifies the highest-ranking subset $\mathcal{A}_{top}$ for ground-truth evaluation.
- Tri-level Update (Lines 23-27): The selected $\mathcal{A}_{top}$ undergo actual evaluation to obtain their ground-truth performance values $\mathcal{J}_{gt}$, which are archived into the history buffer $\mathcal{D}_{hist}$. This triggers a synchronized update: 1) updating $\mathcal{M}_{state}$ with the new candidates $\mathcal{A}_{top}$ to refresh Atlas states; 2) fine-tuning the evaluator parameters $\phi$ using the accumulated experience $\mathcal{D}_{hist}$; and 3) incrementing the iteration $t$ to complete the feedback loop.

### 3.2. Semantic Agent

In the SAGE-NAS framework, the Semantic Agent $\mathcal{A}_\pi$ leverages a Large Language Model (LLM) to transform blind mutation into a context-aware process driven by language priors. To efficiently navigate the high-dimensional search space, we decompose the agent's behavior into four

**Algorithm 1** SAGE-NAS Optimization Procedure

---

**Input:** Space $\mathcal{S}$, Agent $\mathcal{A}_\pi$, Evaluator $\mathcal{E}_\phi$, Max Generations $T_{max}$, Number of Candidates $N$.

**Output:** The architecture $\alpha^*$ with the highest ground-truth performance.

1: **_Phase 0: Initialization_**
2: $\mathcal{P}_{pool} \leftarrow \mathcal{A}_\pi.\text{ZeroShot}(\mathcal{S})$.
3: Behavioral Atlas $\mathcal{M}_{state} \leftarrow \text{Populate}(\mathcal{P}_{pool})$;
4: $\mathcal{P}_0 \leftarrow \text{SelectMaxDispersion}(\mathcal{M}_{state})$.
5: $\mathcal{J}_{gt} \leftarrow \text{Evaluate}(\mathcal{P}_0)$; $\mathcal{D}_{hist} \leftarrow \{(\mathcal{P}_0, \mathcal{J}_{gt})\}$.
6: $\phi \leftarrow \text{Train}(\phi, \mathcal{D}_{hist})$.
7: $t \leftarrow 0$
8: **while** $t < T_{max}$ **do**
9:    **_Phase 1: Context-Guided Semantic Generation_**
10:    $\Omega_t, \mathcal{P}_{elite} \leftarrow \text{Analyze}(\mathcal{M}_{state})$.
11:    $p_{sched} \leftarrow \text{Schedule}(\Omega_t, t)$.    // See Eq. 8
12:    $\mathcal{A}_{cand} \leftarrow \emptyset$.
13:    **for** $i = 1$ to $N$ **do**
14:       $\pi_i \leftarrow \text{SampleSubPolicy}(p_{sched})$.
15:       $\mathcal{F}_i \leftarrow \text{Translate}(\mathcal{M}_{state}, \pi_i, \mathcal{P}_{elite})$.   //Sec. 3.4
16:       $\mathcal{C}_i \leftarrow \text{ConstructPrompt}(\pi_i, \mathcal{F}_i)$.
17:       $\alpha_{new} \leftarrow \mathcal{A}_\pi.\text{Generate}(\mathcal{C}_i)$.   //Sec. 3.2
18:       $\mathcal{A}_{cand} \leftarrow \mathcal{A}_{cand} \cup \{\alpha_{new}\}$.
19:    **end for**
20:    **_Phase 2: Dual-Modality Scoring & Selection_**
21:    $\mathcal{S}_{cand} \leftarrow \mathcal{E}_\phi(\mathcal{A}_{cand})$.   //Sec. 3.3
22:    $\mathcal{A}_{top} \leftarrow \text{SelectTopK}(\mathcal{A}_{cand}, \mathcal{S}_{cand})$.
23:    **_Phase 3: Tri-level State Update_**
24:    $\mathcal{J}_{gt} \leftarrow \text{Evaluate}(\mathcal{A}_{top})$.
25:    $\mathcal{D}_{hist} \leftarrow \mathcal{D}_{hist} \cup \{(\mathcal{A}_{top}, \mathcal{J}_{gt})\}$.
26:    $\mathcal{M}_{state} \leftarrow \text{UpdateAtlas}(\mathcal{M}_{state}, \mathcal{A}_{top}, \mathcal{J}_{gt})$.
27:    $\phi \leftarrow \text{FineTune}(\phi, \mathcal{D}_{hist})$; $t \leftarrow t + 1$.
28: **end while**
29: **return** $\alpha^* = \arg\max_{\alpha \in \mathcal{M}_{state}} \mathcal{J}_{gt}(\alpha)$

---

complementary sub-policies. Crucially, sub-policy execution is regulated by the adaptive scheduling probability $p_{sched}$ derived from the Behavioral Atlas. For every candidate in the generation batch, the agent stochastically samples a specific sub-policy $\pi_i$ based on $p_{sched}$. Conditioned on prompt directives constructed for the sampled $\pi_i \in \{\pi_{local}, \pi_{global}, \pi_{cross}, \pi_{zero}\}$, the LLM executes the corresponding evolutionary operator to directly generate a candidate architecture (refer to Appendix D for details):

**Exploitation**: Topology-Preserving Refinement ($\pi_{local}$). Focusing on elite regions, this strategy instructs the LLM to act as an architecture refiner: freezing the macroscopic topology while fine-tuning microscopic operators. Mathematically, this acts as a local neighborhood search within the discrete search space. Formally, $\alpha_{new}$ is evolved via topology-preserving conditional generation based on $\alpha_{parent}$:

$$\alpha_{new} \sim P_{local}(\cdot \mid \alpha_{parent}, \mathcal{C}_{freeze}, \mathcal{F}_{local}) \quad (2)$$

where $\mathcal{C}_{freeze}$ denotes the prompt constraint enforcing Directed Acyclic Graph (DAG) skeleton invariance. Crucially, $\mathcal{F}_{local}$ represents the Elite Feedback synthesized by the Geometric-Semantic Translator(details in Sec. 3.4). This ensures that the offspring retains the superior topological

structure of $\alpha_{parent}$ while being guided by the collective wisdom derived from these high-performing clusters.

**Exploration**: Semantic-Driven Macroscopic Topological Mutation ($\pi_{global}$). Targeting local stagnation, this strategy employs a semantic intervention mechanism to replace blind stochastic mutations. By constructing topological instructions $\mathcal{I}_{global}$, the system directs the LLM to perform a "semantic jump," such as "disrupting existing connection patterns" or altering connectivity states. This leverages the LLM's semantic priors to break through performance bottlenecks, forcing the search to escape local attractors. The generation is modeled as instruction-guided conditional generation on the targeted parent $\alpha_{parent}$:

$$\alpha_{new} \sim P_{global}(\cdot \mid \alpha_{parent}, \mathcal{I}_{global}) \quad (3)$$

where $\mathcal{I}_{global}$ encapsulates natural language directives for mandatory macroscopic structural alterations, ensuring the offspring diverges topologically from the parent to expand the search scope.

**Crossover**: Chain-of-Thought Compositional Synthesis ($\pi_{cross}$). Traditional NAS crossover operators often lead to structural breakage. We reframe this process as a LLM-driven Chain-of-Thought (CoT) synthesis strategy (Wei et al., 2022). Unlike simple conditional sampling, our strategy comprises: 1) Extraction ($\varphi$): analyzing the core strengths of parents; 2) Synthesis ($\mathcal{S}$): planning a logic for semantic fusion; and 3) Derivation ($\mathcal{D}$): generating the valid architecture. This compositional process is formalized as:

$$\alpha_{new} = \mathcal{D}\big(\mathcal{S}(\varphi(\alpha_A), \varphi(\alpha_B))\big) \quad (4)$$

This semantic-aware crossover ensures grammatical validity while achieving "synergistic" combination to effectively fuse parental strengths and transcend performance limits.

**Novelty-Seeking**: Principle-Driven Zero-Shot Generation ($\pi_{zero}$). For unexplored regions in the Behavioral Atlas inaccessible to mutation-based strategies, $\mathcal{A}_\pi$ switches to the Zero-Shot mode. This strategy decouples from the current population and generates novel architectures based on a set of abstract Design Principles $\mathcal{D}_{principle}$ provided in the Context. Formally, $\alpha_{new}$ is generated from the principle-conditioned distribution:

$$\alpha_{new} \sim P_{zero}(\cdot \mid \mathcal{D}_{principle}, \mathcal{F}_{zero}) \quad (5)$$

The core function of $\pi_{zero}$ is to inject high-potential patterns conforming to $\mathcal{D}_{principle}$ to efficiently populate sparse regions and prevent mode collapse. $\mathcal{F}_{zero}$ represents the Sparsity-Driven Feedback synthesized by the Geometric-Semantic Translator.

### 3.3. Graph-based Evaluator

To provide rapid and accurate performance estimation for the architectures, we propose the Dual-Modality Graph Eval-

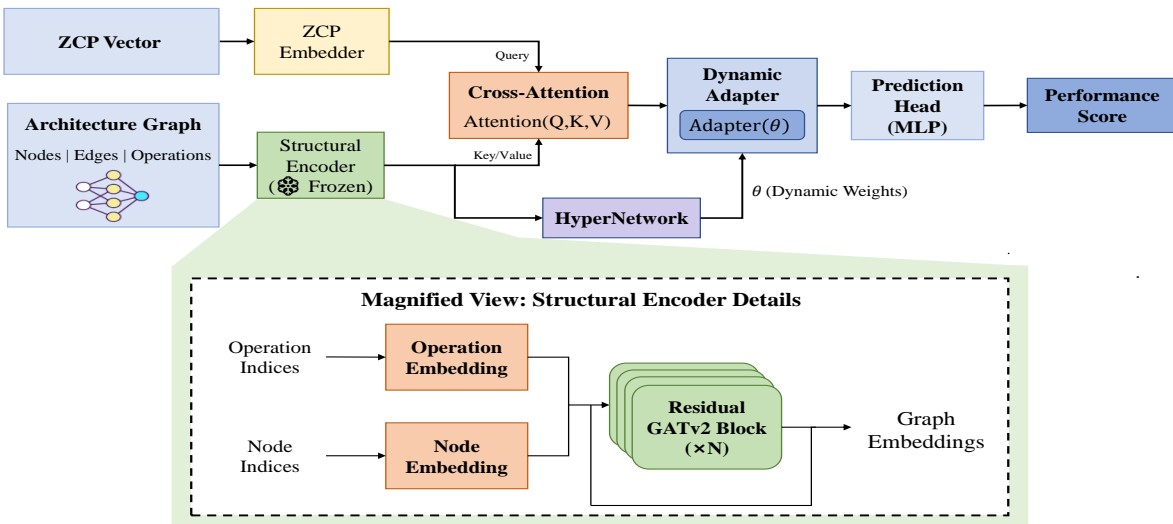

*Figure 2.* The Graph-based Evaluator. The architecture integrates topological structures and physical proxies (ZCP) through three core phases: multi-modal feature extraction via a frozen Structural Encoder, physics-aware fusion using Cross-Attention, and instance-specific refinement driven by a HyperNetwork-guided Dynamic Adapter for precise prediction.

uator ($\mathcal{E}_\phi$). Since obtaining ground-truth labels requires computationally expensive full training, we strictly limit the supervision budget, necessitating an evaluator capable of learning in a data-scarce few-shot regime. Accordingly, this module integrates a "Decoupled Structural Learning" strategy and a "Dual-Modality Fusion" mechanism to predict accurate fitness scores (as in Figure 2).

**Dual-Modality Input Representation.** The model processes two complementary modalities: the structural skeleton formulated as a DAG $\mathcal{G}$ and a physical prior vector $\mathbf{z}_{phy} \in \mathbb{R}^6$. This vector aggregates six complementary ZCPs—NWOT, Jacov, SynFlow, GraSP, L2-Norm, and Params—to comprehensively encode architecture expressivity, gradient dynamics, and spatial complexity. Detailed mathematical formulations for this ZCP suite are provided in Appendix B.3. The DAG $\mathcal{G}$ acts as the blueprint of the neural architecture. It explicitly encodes node configurations, directed edges, and operation types, providing the raw topological definitions necessary for the evaluator.

**Self-Supervised Topological Pre-training.** The proposed "Decoupled Structural Learning" strategy is realized by detaching generic feature extraction from the task-specific fitness estimation. Specifically, we employ a GATv2 backbone (Velickovic et al., 2018; Brody et al., 2022) as the structural encoder (details in Appendix A.2.2). To acquire generic topological priors without relying on expensive ground-truth labels, we pre-train the encoder offline via composite self-supervision (detailed in Appendix B.1). By freezing this backbone during search, we provide stable and generalized structural fingerprints, effectively preventing representation collapse while facilitating the efficient optimization of downstream predictor modules. An abla-

tion study validating the necessity of this self-supervised pre-training is detailed in Appendix C.6.

**Physically-Modulated Cross-Attention.** Instead of simple concatenation, we employ a Physically-Modulated Mechanism to fuse the modalities. First, the physical prior vector $\mathbf{z}_{phy}$ is projected into a latent embedding space via a ZCP Embedder (denoted as $\phi_{zcp}$) to serve as the Query $Q \in \mathbb{R}^{1 \times d}$. Simultaneously, the node-level embeddings $H_{topo} \in \mathbb{R}^{N \times d}$—yielded by the frozen Structural Encoder—are utilized as the Keys ($K$) and Values ($V$). This process leverages the attention mechanism (Vaswani et al., 2017), where the physical priors dynamically re-weight the importance of different topological regions:

$$\text{Attn}(Q, K, V) = \text{Softmax}\left(\frac{QK^T}{\sqrt{d}}\right) V \tag{6}$$
$$\text{where } Q = \phi_{zcp}(\mathbf{z}_{phy}), \quad K, V = H_{topo}$$

This mechanism allows the evaluator to adaptively attend to critical nodes and local substructures that correlate most strongly with the architecture's physical attributes (e.g., trainability), effectively calibrating the prediction bias that might arise from purely topological analysis.

**HyperNetwork-Driven Instance Adaptation.** Addressing structural heterogeneity, we introduce a HyperNetwork $\Psi$ for instance-level adaptation. $\Psi$ is a meta-learner mapping "graph space" to "parameter space": it generates instance-specific weights $\theta_i$ for the Dynamic Adapter based on the global topological fingerprint $h_{\mathcal{G}}^{(i)}$. To obtain a size-invariant representation, we employ global mean pooling:

$$\theta_i = \Psi(h_{\mathcal{G}}^{(i)}), \quad h_{\mathcal{G}}^{(i)} = \frac{1}{N}\sum_{j=1}^{N} H_{topo}^{(i)}[j] \tag{7}$$

where $N$ represents the number of nodes and $H_{topo}^{(i)}[j]$ denotes the feature vector of the $j$-th node. By injecting these dynamic weights $\theta_i$ into the adapter, the evaluator achieves highly adaptive refinement for each specific architecture instance before the final prediction head. As demonstrated in Appendix C.6, this dynamic adaptation effectively acts as a structural regularizer to prevent overfitting in data-scarce regimes, significantly outperforming traditional static adapters.

To optimize this ranking capability, we employ a compound loss function integrating adaptive pairwise hinge loss, distribution alignment loss, and top-k regression loss, as detailed in Appendix B.2.

### 3.4. Behavioral Atlas

SAGE-NAS incorporates the Behavioral Atlas ($\mathcal{M}_{state}$) as a central scheduler to maintain a diverse population of elites via Sparsity-Driven Exploration. Unlike traditional evolutionary algorithms focusing solely on fitness, this module leverages Behavioral State Perception to analyze the search state, bridging physical distributions and semantic planning. To manage the high-dimensional space, we map architectures to a low-dimensional behavioral space and utilize Centroidal Voronoi Tessellation (CVT) (Vassiliades et al., 2017) to discretize the space into non-overlapping niches (configuration details in Table 7). We designate SynFlow and NWOT as the universal axes to encode trainability and expressivity, respectively, while the adaptive resource axis employs FLOPs for fixed-skeleton spaces (e.g., NAS-Bench-201) or Params for variable-topology spaces (e.g., DARTS). Formally, $\mathcal{M}_{state}$ is constructed within this complementary *[Resource, Trainability, Expressivity]* metric space to effectively prevent mode collapse. Visualizations and empirical ablations justifying the selection of these specific axes are provided in Appendix C.3 and Appendix C.7, respectively.

**Sparsity-Driven Feedback.** To bridge the modality gap between the continuous Behavioral Atlas and discrete symbolic reasoning, the system employs a *Geometric-Semantic Translator* (detailed in Appendix D.3). When the Novelty-Seeking mode ($\pi_{zero}$) is sampled, the translator first identifies the targeted unexplored regions. Subsequently, it executes a Geometric Gap Analysis by calculating numerical deviations between the geometric boundaries of these targeted unexplored regions and the nearest neighbor anchors. These state differences are deterministically mapped into semantic navigational descriptors. This generates the sparsity feedback $\mathcal{F}_{zero}$, acting as a navigational compass to guide $\mathcal{A}_\pi$ precisely into these target voids.

**Elite-Driven Feedback.** Conversely, when the Exploitation strategy ($\pi_{local}$) is activated, the system synthesizes feedback to focus the search within high-performing clusters.

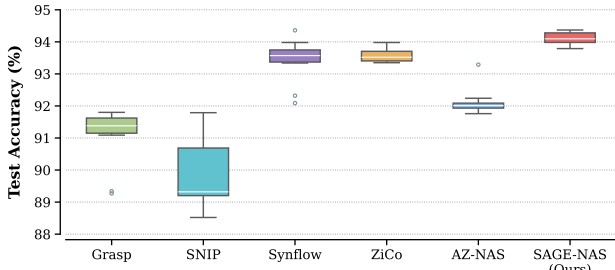

*Figure 3.* Test accuracy distribution of the top-10 candidate architectures on NAS-Bench-201 (CIFAR-10). The box plot compares the quality and stability of SAGE-NAS against zero-cost proxies. The median accuracy is indicated by the white line.

The translator identifies the Top-K elite architectures $\mathcal{P}_{elite}$ to explicitly define the target elite regions within the atlas. Simultaneously, it initiates a Chain-of-Induction process, leveraging the LLM to distill common topological patterns from $\mathcal{P}_{elite}$ into abstract Design Principles $\mathcal{D}_{principle}$. Consequently, the generated elite feedback $\mathcal{F}_{local}$ explicitly integrates the geometric boundaries of these elite regions with the Design Principles, guiding the agent to refine architectures within the target elite regions.

**Adaptive Spatio-Temporal Strategy Scheduling.** To synergistically optimize global discovery and local refinement, we categorize the agent's behaviors into two distinct sets: the exploration set $\Psi_{explore} = \{\pi_{zero}, \pi_{global}\}$ and the exploitation set $\Psi_{exploit} = \{\pi_{local}, \pi_{cross}\}$. We introduce a Spatio-Temporal Gating Mechanism to determine the scheduling probability $p_{sched}(\pi \in \Psi_{explore})$. This probability is computed via a state-aware annealing formulation (detailed definitions for $\lambda(t)$, $\Omega_t$, and $\beta_t$ in Appendix B.5):

$$p_{sched}(\pi \in \Psi_{explore}) = \underbrace{\lambda(t)}_{\text{Temporal Decay}} + \underbrace{\beta_t \cdot \left(1 - \Omega_t(\mathcal{M}_{state})\right)}_{\text{Sparsity Compensation}}$$

(8)

where $\lambda(t)$ is a time-dependent decay factor governing the baseline decay trajectory, and $\Omega_t \in [0, 1]$ represents the current atlas coverage rate. The dynamic coefficient $\beta_t$ serves as a stagnation trigger: it activates the sparsity term $(1 - \Omega_t)$ only when the atlas growth stalls ($\Delta\Omega_t = 0$), thereby re-igniting exploration to break search stagnation. Finally, we employ a uniform allocation strategy within each behavioral set, where the complementary probability is given by $p_{sched}(\pi \in \Psi_{exploit}) = 1 - p_{sched}(\pi \in \Psi_{explore})$.

## 4. Experiments

We evaluate SAGE-NAS across four benchmarks presenting distinct challenges: NAS-Bench-101 (Ying et al., 2019) and NAS-Bench-201 (Dong & Yang, 2020) offer ground-truth mappings for verifying global optimality; DARTS (Liu et al., 2019) tests scalability within a large-scale space ($10^{18}$ candidates); and TransNAS-Bench-101 (Duan et al., 2021) assesses cross-task generalization. We employ Qwen-

*Table 1.* Performance comparison on NAS-Bench-201 (CIFAR-10/100, ImageNet-16-120) and NAS-Bench-101 (CIFAR-10). We report the test accuracy (%) in terms of mean ± standard deviation over 5 independent runs. The row "Optimal" represents the theoretical upper bound of performance in the search space. **Bold** and underlined values indicate the best and second-best results, respectively.

| Algorithm | NAS-Bench-201 (Test Acc %) | | | NAS-Bench-101 | Method |
|---|---|---|---|---|---|
| | CIFAR-10 | CIFAR-100 | ImageNet-16 | CIFAR-10 (Test Acc %) | |
| DARTS (2nd) (Liu et al., 2019) | 54.30 ± 0.00 | 15.61 ± 0.00 | 16.32 ± 0.00 | - | gradient |
| Snip (Abdelfattah et al., 2021) | 86.63 ± 3.89 | 68.88 ± 1.60 | 15.13 ± 10.86 | 85.41 ± 1.38 | training-free |
| Grasp (Abdelfattah et al., 2021) | 88.01 ± 3.14 | 61.43 ± 7.04 | 30.85 ± 5.66 | 87.08 ± 1.40 | training-free |
| Synflow (Abdelfattah et al., 2021) | 93.67 ± 0.39 | 71.70 ± 0.94 | 43.39 ± 3.21 | 89.93 ± 2.97 | training-free |
| ZiCo (Li et al., 2023) | 93.69 ± 0.07 | 70.63 ± 1.08 | 37.18 ± 3.17 | 92.64 ± 0.99 | training-free |
| AZ-NAS (Lee & Ham, 2024) | 93.49 ± 0.30 | 70.33 ± 1.16 | 44.34 ± 1.26 | 92.01 ± 0.85 | training-free |
| SWAP (Peng et al., 2024) | 90.48 ± 0.94 | 67.13 ± 1.83 | 35.40 ± 3.96 | 90.51 ± 2.08 | training-free |
| RoBoT (He et al., 2024b) | 94.36 ± 0.00 | **73.51 ± 0.00** | 46.34 ± 0.00 | - | hybrid |
| PO-NAS (Lin & Luo, 2026) | 94.12 ± 0.22 | **73.51 ± 0.00** | 46.71 ± 0.12 | - | hybrid |
| LLMatic (Nasir et al., 2024) | 94.26 ± 0.13 | 71.62 ± 1.73 | 45.87 ± 0.96 | - | LLM |
| RZ-NAS (Ji et al., 2025) | 94.24 ± 0.12 | 73.30 ± 0.21 | 46.24 ± 0.23 | - | LLM |
| **SAGE-NAS (Ours)** | **94.37 ± 0.00** | **73.51 ± 0.00** | **47.05 ± 0.31** | **93.73 ± 0.13** | LLM |
| Optimal | 94.37 | 73.51 | 47.31 | 94.32 | - |

*Table 2.* Comparison on the DARTS search space using CIFAR-10/CIFAR-100 and ImageNet. We report Test Error (%) for CIFAR datasets (mean ± standard deviation over 5 independent runs for CIFAR-10) and Top-1/Top-5 Error (%) for ImageNet. "Params" (M) refers to the model size under the ImageNet setting, while "Cost" (GPU Days) denotes the search cost evaluated on an Nvidia 1080Ti; for SAGE-NAS, this cost explicitly includes the latency incurred by LLM API calls. **Bold** and underlined values denote the best and second-best results, respectively. "-" indicates that the result is unavailable or not reported.

| Algorithm | CIFAR-10 | CIFAR-100 | ImageNet | | | | Method |
|---|---|---|---|---|---|---|---|
| | Test Err (%) | Test Err (%) | Top-1 Err | Top-5 Err | Params (M) | Cost (GPU Days) | |
| DARTS (2nd) (Liu et al., 2019) | 2.76 ± 0.09 | 17.52 | 26.7 | 8.7 | 4.7 | 4 | gradient |
| PC-DARTS (Xu et al., 2020) | 2.57 ± 0.07 | 17.11 | 24.2 | 7.3 | 5.3 | 3.7 | gradient |
| IS-DARTS (He et al., 2024a) | 2.56 ± 0.04 | - | 24.1 | 7.1 | 6.4 | 0.42 | gradient |
| AmoebaNet-A (Real et al., 2019) | 2.55 ± 0.05 | 16.82 | 25.5 | 7.6 | 6.4 | 3150 | evolution |
| TENAS (Chen et al., 2021b) | 2.63 ± 0.06 | - | 24.5 | 7.5 | 5.4 | 0.17 | training-free |
| SWAP (Peng et al., 2024) | 2.48 ± 0.09 | - | 24.0 | 7.6 | 5.8 | 0.006 | training-free |
| RoBoT (He et al., 2024b) | 2.60±0.03 | 16.52 | 24.1 | 7.3 | 5.0 | 0.6 | hybrid |
| PO-NAS (Lin & Luo, 2026) | 2.52±0.03 | 16.35 | 23.9 | 7.1 | 6.3 | 0.64 | hybrid |
| RZ-NAS (Ji et al., 2025) | **2.41 ± 0.13** | 17.49 | - | - | - | - | LLM |
| **SAGE-NAS (Ours)** | 2.42 ± 0.06 | **15.17** | **23.7** | **6.8** | 5.9 | 0.86 | LLM |

Plus (Bai et al., 2023) as the Semantic Agent LLM Backbone. For detailed search space specifications and hyperparameters, refer to Appendix A.1 and Appendix A.2.

## 4.1. Performance Comparison

### 4.1.1. NAS-BENCH-201&101 SEARCH SPACE

We evaluate SAGE-NAS on the NAS-Bench-201 and NAS-Bench-101 benchmarks, with quantitative comparisons summarized in Table 1. SAGE-NAS significantly outperforms training-free baselines and competitive LLM-driven methods across all datasets. Notably, within the NAS-Bench-201 search space, it successfully identifies the theoretical global optimum on both CIFAR-10 and CIFAR-100, while achieving superior accuracy on ImageNet-16-120 (47.05%). Furthermore, on the NAS-Bench-101 benchmark, it achieves a leading accuracy of 93.73%. To further assess search robust-

ness, we visualize the quality distribution of the top-10 candidates on NAS-Bench-201 (CIFAR-10) in Figure 3. In stark contrast to traditional proxies plagued by rank-disordered outliers, SAGE-NAS exhibits a distribution with compact convergence towards the high-accuracy region.

### 4.1.2. DARTS SEARCH SPACE

Table 2 details the performance within the expansive DARTS space. SAGE-NAS achieves a competitive test error of 2.42% on CIFAR-10 and reaches 15.17% (SOTA) on the more challenging CIFAR-100. On the large-scale ImageNet dataset, the model attains a 23.7% Top-1 and 6.8% Top-5 error, consistently outperforming existing counterparts. Crucially, these results are achieved with a search cost of only 0.86 GPU Days, highlighting an exceptional balance between search efficiency and cross-task generalization capability (visualized in Appendix C.5).

*Table 3.* Performance comparison on the TransNAS-Bench-101 (Micro) search space. We report the mean ± standard deviation across 5 independent runs. Implementation details for the baselines (REA, RS, REINFORCE) are provided in Appendix A.2.4. **Bold** and underlined values indicate the best and second-best performance, respectively.

| Method | Object (Acc %) | Scene (Acc %) | Jigsaw (Acc %) | Autoencoder (SSIM) | Normal (SSIM) | Segment (mIoU) | Layout (Neg. L2 Loss) |
|---|---|---|---|---|---|---|---|
| REA | 44.88±0.31 | 54.63±0.18 | 94.73±0.14 | **56.00±0.71** | 56.76±0.33 | 94.56±0.02 | -62.02±0.59 |
| RS | 44.76±0.45 | 54.53±0.19 | 94.61±0.29 | 55.27±0.90 | 56.45±0.25 | 94.52±0.03 | -62.17±0.98 |
| REINFORCE | 44.64±0.38 | 54.49±0.19 | 94.69±0.16 | 55.47±0.72 | 57.06±0.35 | 94.53±0.04 | -61.61±0.91 |
| RoBoT (He et al., 2024b) | **45.59±0.00** | 54.87±0.00 | 94.82±0.06 | 55.42±1.05 | **57.44±0.34** | 94.58±0.00 | -61.16±0.86 |
| PO-NAS (Lin & Luo, 2026) | **45.59±0.00** | 54.90±0.03 | 94.91±0.09 | 55.62±0.41 | 57.08±0.14 | **94.60±0.01** | -61.55±0.75 |
| **SAGE-NAS (Ours)** | **45.59±0.00** | **54.91±0.02** | **95.20±0.15** | 55.45±0.99 | 57.26±0.07 | 94.58±0.03 | **-60.84±1.04** |
| Optimal | 45.59 | 54.94 | 95.37 | 57.72 | 58.73 | 94.61 | -60.10 |

*Table 4.* Performance comparison within the AutoFormer search space on ImageNet. "-" indicates that the search cost is not applicable or not reported. **Bold** values indicate the best performance.

| Algorithm | Param (M) | Top-1 (%) | GPU Days |
|---|---|---|---|
| DeiT-Ti (Touvron et al., 2021) | 5.7 | 72.2 | - |
| TNT-Ti (Han et al., 2021) | 6.1 | 73.9 | - |
| ViT-Ti (Dosovitskiy et al., 2021) | 5.7 | 74.5 | - |
| PVT-Tiny (Wang et al., 2021) | 13.2 | 75.1 | - |
| ViTAS-C (Su et al., 2022) | 5.6 | 74.7 | 32 |
| AutoFormer-Ti (Chen et al., 2021a) | 5.7 | 74.7 | 24 |
| TF-TAS-Ti (Zhou et al., 2022) | 5.9 | 75.3 | 0.5 |
| Auto-Prox (Wei et al., 2024) | 6.4 | 75.6 | 0.1 |
| ParZC (Dong et al., 2025b) | 6.1 | 75.5 | 0.05 |
| **SAGE-NAS (Ours)** | 7.4 | **75.8** | 0.47 |

### 4.1.3. TRANSNAS-BENCH-101–MICRO SEARCH SPACE

We evaluate the cross-task generalization of SAGE-NAS on the TransNAS-Bench-101 (Micro) search space to verify its potential across multimodal vision tasks, as summarized in Table 3. SAGE-NAS exhibits exceptional architectural adaptability, achieving state-of-the-art performance across four distinct tasks: Object, Scene, Jigsaw, and Layout. Furthermore, even in challenging pixel-level prediction scenarios, such as Normal Estimation and Semantic Segmentation, SAGE-NAS consistently ranks within the top two. These results robustly validate the method's capability to effectively adapt topological structures to the differentiated requirements of diverse vision tasks, ranging from high-level classification to low-level geometric understanding.

### 4.1.4. AUTOFORMER SEARCH SPACE

Table 4 details the performance comparison within the AutoFormer search space on ImageNet. SAGE-NAS achieves a state-of-the-art Top-1 accuracy of 75.8%, outperforming both hand-crafted Vision Transformers and competitive NAS baselines. Notably, although its parameter size (7.4M) is slightly larger than some extremely lightweight counterparts, SAGE-NAS completes the search process in only 0.47 GPU Days, achieving an exceptional balance between search efficiency and final representational capacity. These results robustly validate the capability of the proposed method to rapidly and accurately identify high-performance network topologies within transformer-based macro spaces.

*Table 5.* Ablation study of SAGE-NAS on NAS-Bench-201 across 5 runs. "w/o" denotes variants with specific modules removed.

| Method | Test Accuracy (%) | | |
|---|---|---|---|
| | CIFAR-10 | CIFAR-100 | ImageNet-16 |
| w/o Semantic Agent | 94.04 ± 0.31 | 71.40 ± 0.56 | 45.70 ± 0.24 |
| w/o Graph Evaluator | 94.29 ± 0.06 | 73.14 ± 0.04 | 45.87 ± 0.86 |
| w/o Behavioral Atlas | 94.11 ± 0.33 | 73.09 ± 0.00 | 46.07 ± 0.49 |
| **SAGE-NAS (Ours)** | **94.37 ± 0.00** | **73.51 ± 0.00** | **47.05 ± 0.31** |

*Table 6.* Influence of LLM Capability. Comparison of search performance and efficiency across different LLM backbones. Experiments are performed on CIFAR-10 within the DARTS search space. "Cost" denotes the total API expenditure in US Dollars.

| LLM Backbone | Test Accuracy (%) | | | | Efficiency |
|---|---|---|---|---|---|
| | Run 1 | Run 2 | Run 3 | Avg. | API Cost ($) |
| GPT-5.1 | 97.55 | 97.59 | 97.37 | 97.50 | 8.36 |
| Claude-haiku-4.5 | 97.36 | 97.43 | 97.44 | 97.41 | 13.73 |
| Gemini-3-flash-preview | 97.50 | 97.36 | 97.22 | 97.36 | 8.01 |
| Qwen-plus | 97.58 | 97.52 | 97.46 | 97.52 | 2.01 |

### 4.2. Ablation studies

#### 4.2.1. IMPACT OF CORE COMPONENTS

To rigorously validate the contribution of each module, we conduct component-wise ablation studies on NAS-Bench-201 (details in Appendix C.1), with quantitative results in Table 5. As observed, the SAGE-NAS framework consistently outperforms all incomplete variants. Specifically, removing the Semantic Agent ($\mathcal{A}_\pi$) results in the most substantial performance degradation, confirming that LLM-based in-context reasoning provides a decisive advantage over blind stochastic search. Furthermore, excluding the Behavioral Atlas ($\mathcal{M}_{state}$) leads to statistically significant accuracy drops accompanied by increased variance, underscoring its indispensable role in preventing mode collapse. Finally, the absence of the Dual-Modality Graph Evaluator ($\mathcal{E}_\phi$) exposes the system to semantic-physical misalignment, hindering the precise identification of optimal architectures. Evolutionary trajectories are visualized in Appendix C.4.

### 4.2.2. INFLUENCE OF LLM CAPABILITY

To investigate the impact of the inference engine on search efficacy and economic cost, we compare four mainstream LLMs within the DARTS search space on CIFAR-10 (see Table 6). Experimental results reveal two key findings: (1) Framework Robustness: SAGE-NAS exhibits strong adaptability to the backbone model. The maximum difference in average accuracy across different models is only 0.16%, demonstrating the algorithm's insensitivity to the underlying reasoning core. (2) Optimal Backbone Selection: Qwen-plus achieves the optimal efficiency balance with the highest average accuracy of 97.52% and the lowest search cost of $2.01. Compared to the second-best performing GPT-5.1, Qwen-plus slightly improves accuracy while reducing API costs by approximately $4\times$ ($8.36 vs. $2.01). Based on this significant cost-effectiveness advantage, we establish Qwen-plus as the foundational LLM for this study.

## 5. Conclusion

In this work, we present SAGE-NAS, a synergistic evolutionary framework to address the "Semantic-Physical Misalignment" gap in neural architecture search. By establishing a closed-loop optimization mechanism, we deeply integrate the high-level semantic planning capabilities of Large Language Models with the low-level physical sensitivity of Zero-Cost Proxies. Specifically, the framework orchestrates an LLM-Based Semantic Agent to execute context-guided sub-policies for robust generation, employs a Dual-Modality Graph Evaluator to inject physical priors into topological embeddings for fast and accurate evaluation, and leverages a State-Aware Behavioral Atlas to guide sparsity-driven exploration. This synergy effectively mitigates mode collapse and achieves a dynamic equilibrium between exploration and exploitation. Empirically, SAGE-NAS achieves state-of-the-art performance across mainstream benchmarks with exceptional search efficiency, exhibiting superior robustness and cross-task generalization.

## 6. Limitations and Future Work

Despite its promising results, our framework exhibits certain limitations that outline clear avenues for future research. First, regarding efficiency trade-offs, our Dual-Modality Graph Evaluator requires a one-time topological pre-training for each unique search space, followed by online few-shot adaptation during each independent search process. Consequently, for strictly resource-constrained scenarios requiring instant, "plug-and-play" scoring without any warm-up cost, standalone Zero-Cost Proxies remain computationally advantageous. Second, extending SAGE-NAS to highly complex domains—such as true hierarchical search spaces—presents scalability challenges that may strain both the cur-

rent flat graph evaluator and the context capacity of the LLM.

To address these boundaries, our future work will focus on three primary directions: (1) developing hierarchical graph encoders to natively capture multi-level architectural dependencies and prevent context explosion; (2) transitioning the Behavioral Atlas to a task-adaptive mechanism via bi-objective optimization, which dynamically aligns ZCP combinations with specific task characteristics; and (3) exploring NAS-specific instruction-tuning for Small Language Models (SLMs). This final direction aims to bridge the performance gap with massive closed-source APIs, ultimately enabling fully offline, high-performance automated architecture design.

## Acknowledgements

This work was supported by the Shenzhen Key Laboratory Fund under Grant SYSPG20241211174032004, and the Scientific Research and Development Foundations of Shenzhen under Grant JCYJ20220818100005011.

## Impact Statement

This paper presents work whose goal is to advance the field of Machine Learning. There are many potential societal consequences of our work, none which we feel must be specifically highlighted here.

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

## Appendix A. Search Space Definitions & Training Configurations

This section details the static configurations of the search spaces and the dynamic hyperparameters used during the architecture search and model training experiments.

### Appendix A.1. Search Space Specifications

We evaluate SAGE-NAS across four distinct benchmarks, each possessing unique structural characteristics and complexity scales. The specific topological constraints and primitive operation sets are defined as follows:

- **NAS-Bench-201:** This search space adopts a cell-based structure, where the macro-skeleton is formed by stacking 17 cells. Each cell is modeled as a Directed Acyclic Graph (DAG) with 4 nodes. Since every pair of nodes allows for a connection, each cell contains exactly 6 edges. The operation for each edge is selected from a pre-defined candidate set $\mathcal{O}_{201}$ containing 5 primitives:

$$\mathcal{O}_{201} = \{\texttt{zero}, \texttt{skip\_connect}, \texttt{conv\_1x1}, \texttt{conv\_3x3}, \texttt{avg\_pool\_3x3}\}$$

  Consequently, the search space encompasses a total of $5^6 = 15,625$ unique neural architectures. This benchmark provides complete ground-truth performance queries on CIFAR-10, CIFAR-100, and ImageNet-16-120, serving as an ideal platform for diagnosing search trajectories and verifying global optimality.

- **NAS-Bench-101:** This represents a graph-based search space with loose topological constraints. Each cell is represented as a DAG allowing up to 7 vertices and 9 edges. Unlike the densely connected structure of NAS-Bench-201, this space exhibits greater topological flexibility. The first node serves as the fixed input and the last as the fixed output, while the operations for intermediate nodes are selected from a set of 3 primitives:

$$\mathcal{O}_{101} = \{\texttt{conv\_1x1}, \texttt{conv\_3x3}, \texttt{max\_pool\_3x3}\}$$

  The space contains 423,624 unique neural architectures, primarily utilized to evaluate algorithmic performance within a space characterized by complex topological variations.

- **DARTS:** This benchmark represents an ultra-large-scale search space with high combinatorial complexity (approximately $10^{18}$ candidates). The search process aims to design two types of convolutional cells: a normal cell and a reduction cell. Each cell contains 7 nodes, consisting of 2 input nodes, 4 intermediate computation nodes, and 1 output node. Each intermediate node is connected to two preceding nodes via edges. The candidate operation set $\mathcal{O}_{darts}$ includes 8 primitives, covering convolutions with different receptive fields and pooling operations:

$$\mathcal{O}_{darts} = \{\texttt{sep\_3x3}, \texttt{sep\_5x5}, \texttt{dil\_3x3}, \texttt{dil\_5x5}, \texttt{max\_3x3}, \texttt{avg\_3x3}, \texttt{skip}, \texttt{zero}\}$$

  This space is employed to verify the scalability of SAGE-NAS in handling complex structures, such as dilated and separable convolutions.

- **TransNAS-Bench-101 (Micro):** To assess cross-task generalization capabilities, we adopt the Micro search space from TransNAS-Bench-101. This space is topologically isomorphic to NAS-Bench-201, featuring the same structure of 4 nodes and 6 edges per cell. However, to accommodate diverse vision tasks, the operation set excludes average pooling and is reduced to 4 primitives:

$$\mathcal{O}_{trans} = \{\texttt{zero}, \texttt{skip\_connect}, \texttt{conv\_1x1}, \texttt{conv\_3x3}\}$$

  This results in a search space containing $4^6 = 4,096$ architectures. We evaluate these architectures across 7 distinct vision tasks, including object classification, scene classification, semantic segmentation, autoencoding, surface normal estimation, jigsaw puzzle solving, and room layout estimation.

### Appendix A.2. Hyperparameter Settings

To facilitate reproducibility, we provide a comprehensive breakdown of the hyperparameters used in SAGE-NAS. We categorize these configurations into three parts: evolutionary search settings, semantic agent configurations, and search space-specific training protocols (e.g., DARTS).

APPENDIX A.2.1. EVOLUTIONARY SEARCH & AGENT CONFIGURATION

Table 7 details the default parameters for the evolutionary engine and the LLM-based agent. The search process is driven by a population-based strategy with a fixed random seed for fairness.

*Table 7.* Default hyperparameters for SAGE-NAS. We adopt a three-pillar configuration: Evolutionary Search (Left), Behavioral Atlas (Center), and Semantic Agent (Right).

| 1. Evolutionary Search | | 2. Behavioral Atlas (CVT) | | 3. Semantic Agent | |
|---|---|---|---|---|---|
| **Parameter** | **Value** | **Parameter** | **Value** | **Parameter** | **Value** |
| Init. Pool ($\mathcal{P}_{pool}$) | 50 | Num Niches | 512 | LLM Backbone | Qwen-Plus |
| Init. Pop. Size ($\mathcal{P}_0$) | 24 | CVT Samples | 10,000 | Output Format | JSON Schema |
| Max Gens ($T_{max}$) | 48 | *Axis 1 (Resource)* | Params/FLOPs | Temp. ($\pi_{local}$) | 0.4–0.8 |
| Parent/Gen | 12 | *Axis 2 (Trainability)* | SynFlow | Temp. ($\pi_{cross}$) | 0.7 |
| Offspring/Gen ($N$) | 30 | *Axis 3 (Expressivity)* | NWOT | Temp. ($\pi_{global}$) | 0.6–1.0 |
| True Eval/Gen | 1 or 2[†] | Random Seed | 16 | Temp. ($\pi_{zero}$) | 0.8–1.2 |

[†]*To balance computational cost, we allocate 1 ground-truth evaluation per generation for the extensive DARTS search space, and 2 for the tabular benchmarks (NAS-Bench-101/201, TransNAS-Bench-101 (Micro)).*

APPENDIX A.2.2. GRAPH EVALUATOR CONFIGURATIONS & TRAINING PROTOCOL

The configuration of the Dual-Modality Graph Evaluator encompasses two critical dimensions: the selection of Zero-Cost Proxies (ZCPs) and a decoupled training strategy.

**1. Zero-Cost Proxy Configuration.** To capture physical architectural properties and enable efficient cold-start, we construct the physical feature vector using six complementary training-free metrics: `nwot`, `jacov`, `l2_norm`, `params`, `synflow`, and `grasp`. These proxies provide rich prior knowledge regarding parameter efficiency, gradient flow, and saliency without requiring gradient updates. (Detailed mathematical definitions of these metrics are provided in Appendix B).

**2. Detailed Network Architecture.** To ensure full reproducibility, we explicitly detail the layer configurations and dimensions for the Graph Evaluator ($\mathcal{E}_\phi$) modules based on our implementation. As listed in Table 8, the system employs a deep GATv2 backbone to capture complex topology, coupled with GELU-activated projectors for the dual-modality fusion.

*Table 8.* Detailed Architectural Specifications of the Graph Evaluator components.

| Module | Component | Configuration Details |
|---|---|---|
| **Structural Encoder** | Backbone
Layers ($N$)
Hidden Dim ($d$)
Heads | GATv2
4 (with Residual Connections)
64
4 |
| **ZCP Embedder** | Architecture
Dimensions | MLP (Linear → GELU → Linear)
Input: 6 → Hidden: 32 → Out: 64 |
| **Fusion Module** | Attention
Adapter | Multi-Head Cross-Attention (Heads=4, $d = 64$)
Dynamic BottleNeck (64 → 32 → 64) |

**3. Decoupled Training Protocol.** To balance representation robustness with adaptation flexibility under the few-shot regime, we adopt a decoupled training strategy. The lifecycle is strictly divided into feature learning and performance mapping:

1. Offline Pre-training: Solely the Structural Encoder is optimized using self-supervised objectives (defined in Appendix B) to derive generic topological embeddings. The unlabeled pre-training dataset is tailored to the cardinality of the target search space: for small-scale benchmarks (NAS-Bench-201 and TransNAS-Bench-101 Micro), we utilize all candidate architectures within the search space; for large-scale spaces (NAS-Bench-101 and DARTS), we randomly sample a subset of 10,000 unique unlabeled architectures to ensure efficient coverage.

2. Downstream Adaptation: During the subsequent "Initial Training" and "Online Fine-tuning" stages, the Structural Encoder weights are frozen to provide stable fingerprints. Optimization is applied to all downstream modules, including the ZCP Embedder, Cross-Attention layers, HyperNetwork-based Dynamic Adapter, and the Prediction Head.

This strategy effectively prevents overfitting of the heavy graph backbone while allowing the dual-modality fusion mechanism to rapidly adapt to the search context. Detailed hyperparameters are listed in Table 9.

*Table 9.* Training configurations for the Graph Evaluator. The Structural Encoder is frozen during the search phase, while all subsequent modules are trainable.

| Stage & Target | Hyperparameter | Value |
|---|---|---|
| **1. Offline Pre-training** *(Struct. Encoder Only)* | Epochs | 100 |
| | Batch Size | 128 |
| | Learning Rate | $1 \times 10^{-3}$ |
| | Optimizer | AdamW |
| **2. Initial Training** *(Encoder Frozen)* | Epochs | 40 |
| | Data Source | Initial Pop. ($N = 24$) |
| | Learning Rate | $2 \times 10^{-4}$ |
| **3. Online Fine-tuning** *(Encoder Frozen)* | Update Frequency | Every Generation |
| | Epochs | 20 |
| | Batch Size | 32 |
| | Learning Rate | $2 \times 10^{-4}$ |

APPENDIX A.2.3. DARTS SEARCH SPACE SETTINGS

For the DARTS search space, which requires full training, we adopt distinct strategies for the search phase and the final evaluation (retraining) phase. Table 10 summarizes these configurations.

*Table 10.* Training hyper-parameters for the DARTS search space. We adopt distinct strategies for the search phase and final evaluation on CIFAR and ImageNet.

| Parameter | Search | Retraining Phase (Evaluation) | |
|---|---|---|---|
| | (Proxy) | CIFAR | ImageNet |
| Epochs | 12 | 600 | 250 |
| Batch Size | 128 | 96 | 128 |
| Initial Channels | 16 | 36 | 48 |
| Layers | 8 | 20 | 14 |
| Optimizer | SGD | SGD | SGD |
| Learning Rate (LR) | 0.025 | 0.025 | 0.1 |
| LR Scheduler | Cosine | Cosine | Cosine |
| Momentum | 0.9 | 0.9 | 0.9 |
| Weight Decay | $3 \times 10^{-4}$ | $3 \times 10^{-4}$ | $3 \times 10^{-5}$ |
| Gradient Clipping | 5 | 5 | 5 |
| Cutout Length | - | 16 | - |
| Drop Path Prob. | 0 | 0.2 | 0.2 |
| Auxiliary Weight | 0 | 0.4 | 0.4 |
| Label Smoothing | - | - | 0.1 |

APPENDIX A.2.4. BASELINE IMPLEMENTATION DETAILS

In this section, we elaborate on the implementation protocols for the baseline algorithms used in our comparative study. All baselines are executed under the same search budget constraints to ensure fairness.

**Random Search (RS).** Serving as a fundamental baseline for evaluating search efficiency, Random Search operates by uniformly sampling architectures from the discrete search space without any learning mechanism. In our experiments, we sample architectures independently and select the candidate with the highest validation accuracy. This unbiased exploration provides a crucial lower bound for measuring the effectiveness of guided search strategies.

**Regularized Evolutionary Algorithm (REA).** We implement the aging evolution strategy proposed by Real et al. (Real et al., 2019). Consistent with standard protocols, the population is initialized by randomly sampling architectures using one-third of the total query budget. During the evolutionary process, a tournament selection is performed on the current

population to identify the best parent, which is then mutated to produce a new offspring. Crucially, to maintain population diversity, the oldest individual in the population is removed upon the insertion of the new offspring. For the cell-based search spaces utilized in this work (NAS-Bench-201 and TransNAS-Bench-101-Micro), the mutation operation is defined as randomly modifying the operation type of a single edge within the cell's Directed Acyclic Graph (DAG).

**REINFORCE.** This baseline utilizes the classic policy gradient method (Williams, 1992) to optimize the architecture generation policy, as adopted in early NAS works (Zoph & Le, 2017), where the controller is modeled to maximize the expected validation accuracy. In our implementation, we employ the Adam optimizer with a learning rate of 0.01 to update the policy parameters. Furthermore, to stabilize the gradient estimation and ensure robust convergence, we incorporate a baseline reward mechanism, calculated as the exponential moving average of previous rewards with a momentum factor of 0.9.

## Appendix B. Methodological Details & Mathematical Formulations

This section provides the mathematical formulations and theoretical definitions omitted from the main text for brevity, specifically detailing the pre-training objectives derived from the code implementation and the compound loss functions used for online fine-tuning.

### Appendix B.1. Self-Supervised Pre-training Objectives

To empower the Structural Encoder with generic topological semantics before downstream adaptation, we employ a composite self-supervised objective $\mathcal{L}_{ssl} = \mathcal{L}_{deg} + \mathcal{L}_{op} + \mathcal{L}_{cl}$. The specific formulations are as follows:

**1. Node Degree Regression ($\mathcal{L}_{deg}$).** Inspired by the `GraphDegreePretrainModel` implementation, we replace the computationally expensive adjacency reconstruction with node degree regression to capture topological density. The model predicts the total degree $d_v$ (in-degree + out-degree) for each node $v$. Let $\phi_{deg}(\cdot)$ denote the projector head:

$$\mathcal{L}_{deg} = \frac{1}{|\mathcal{V}|} \sum_{v \in \mathcal{V}} \|\phi_{deg}(\mathbf{h}_v) - d_v\|^2 \tag{9}$$

Minimizing this Mean Squared Error (MSE) forces the node embeddings $\mathbf{h}_v$ to explicitly encode local connectivity patterns.

**2. Edge Operation Reconstruction ($\mathcal{L}_{op}$).** To understand the semantic interaction between components, this task requires the model to infer the operation type $o_{uv} \in \mathcal{O}$ connecting two nodes based solely on their embeddings. Let $\phi_{op}$ be a classifier:

$$\mathcal{L}_{op} = -\frac{1}{|\mathcal{E}|} \sum_{(u,v) \in \mathcal{E}} \sum_{k=1}^{|\mathcal{O}|} \mathbb{I}(o_{uv} = k) \cdot \log\left(\text{softmax}(\phi_{op}([\mathbf{h}_u \| \mathbf{h}_v]))_k\right) \tag{10}$$

where $\|$ denotes concatenation. This ensures the encoder distinguishes between different primitives (e.g., convolution vs. pooling) within the topology.

**3. Graph Contrastive Learning ($\mathcal{L}_{cl}$).** To enhance global robustness against structural perturbations, we apply two stochastic augmentation strategies: random edge dropping (30%) and random operation swapping (20%). Given a graph $\mathcal{G}$, we generate two correlated views $\hat{\mathcal{G}}_i, \hat{\mathcal{G}}_j$ and obtain graph-level embeddings $\mathbf{z}_i, \mathbf{z}_j$ via global pooling. The objective is to minimize the symmetrized NT-Xent loss:

$$\ell_{i,j} = -\log \frac{\exp(\text{sim}(\mathbf{z}_i, \mathbf{z}_j)/\tau)}{\sum_{k=1}^{2B} \mathbb{I}_{[k \neq i]} \exp(\text{sim}(\mathbf{z}_i, \mathbf{z}_k)/\tau)} \tag{11}$$

$$\mathcal{L}_{cl} = \frac{1}{2B} \sum_{i=1}^{B} (\ell_{i,j} + \ell_{j,i}) \tag{12}$$

where $\text{sim}(\cdot)$ denotes the cosine similarity, $B$ is the batch size, and $\tau$ is the temperature parameter set to 0.2 in our experiments.

## Appendix B.2. Online Fine-tuning Objectives

During the search phase, to calibrate the ranking fidelity and adapt to the local search space, we employ a compound loss function:

$$\mathcal{L}_{total} = \gamma \cdot \mathcal{L}_{pairwise} + \delta \cdot \mathcal{L}_{dist} + \epsilon \cdot \mathcal{L}_{top\text{-}k} \tag{13}$$

We set the balancing coefficients to $\gamma = 1.0$, $\delta = 0.5$, and $\epsilon = 0.3$ by default, prioritizing pairwise ranking accuracy while using distribution alignment and top-k regression as auxiliary regularizers.

**1. Adaptive Pairwise Hinge Loss ($\mathcal{L}_{pairwise}$).** Standard hinge loss uses a fixed margin, which causes gradients to vanish when ranking elite architectures with similar performance. We introduce an adaptive margin proportional to the ground-truth accuracy gap. For a pair $(A, B)$ where true accuracy $y_A > y_B$:

$$\mathcal{L}_{pairwise} = \max(0, \alpha \cdot (y_A - y_B) - (s_A - s_B)) \tag{14}$$

where $s_A, s_B$ are predicted scores and $\alpha$ is a scaling factor set to 0.6. This penalizes the model only if the predicted score gap fails to reflect the true performance gap significantly.

**2. Distribution Alignment Loss ($\mathcal{L}_{dist}$).** To prevent "scale distortion" where the model ranks correctly but compresses the score range, we align the statistical distribution of predicted gaps with true gaps. We minimize the distance between the first and second moments of the difference distributions:

$$\mathcal{L}_{dist} = \|\mu_{\Delta s} - \mu_{\Delta y}\|^2 + \|\sigma_{\Delta s} - \sigma_{\Delta y}\|^2 \tag{15}$$

where $\Delta s$ and $\Delta y$ represent the set of pairwise differences in predictions and ground-truths within a batch, respectively.

**3. Top-K Regression Loss ($\mathcal{L}_{top\text{-}k}$).** In NAS, accurately predicting poor-performing architectures is irrelevant. We focus regression solely on the top-performing candidates. Given a batch of size $N$, we sort samples by ground-truth $y$ and select the top $K = \sqrt{N}$ indices:

$$\mathcal{L}_{top\text{-}k} = \frac{1}{K} \sum_{i \in \text{Top-K}} (s_i - y_i)^2 \tag{16}$$

This strategy prevents the model from overfitting to noisy, non-competitive architectures.

## Appendix B.3. Mathematical Definitions of Zero-Cost Proxies

This section provides the rigorous mathematical formulations for the six Zero-Cost Proxies (ZCPs) utilized to construct the physical embedding vector. All metrics are computed on a randomly initialized network with parameters $\theta$. To ensure reproducibility and decouple metric computation from dataset sampling variance, we utilize a single mini-batch of **synthetic data** $\mathcal{B} = \{(\mathbf{x}_i, y_i)\}_{i=1}^N$.

**Data Generation Protocol.** We set the batch size to $N = 64$. The synthetic inputs and targets are generated as follows:

- **Inputs (x):** Images are generated with a resolution of $3 \times 32 \times 32$ (aligned with CIFAR-10 shapes), where each pixel is independently sampled from a standard normal distribution $\mathcal{N}(0, 1)$.
- **Targets ($y$):** Labels are assigned as uniformly random integers sampled from the range $[0, C - 1]$, where $C$ denotes the number of classes (e.g., $C = 10$).

Note that metrics requiring specific input forms (e.g., SynFlow) override this default configuration as detailed below.

**1. NWOT** (Mellor et al., 2021). Neural Architecture Search withOut Training (NWOT) quantifies expressivity via the diversity of activation patterns. Let $\mathbf{c}_i \in \{0, 1\}^M$ denote the binary code vector derived from the ReLU activations of the network for input $\mathbf{x}_i$, where $M$ is the total number of neurons. We compute the kernel matrix $\mathbf{K}_H \in \mathbb{R}^{N \times N}$, where each element $\mathbf{K}_{H,ij} = M - d_H(\mathbf{c}_i, \mathbf{c}_j)$ represents the Hamming similarity. The score is defined as:

$$S_{nwot} = \log \det(\mathbf{K}_H) \tag{17}$$

where $d_H(\cdot, \cdot)$ is the Hamming distance and $\det(\cdot)$ denotes the determinant.

**2. Jacov** (Abdelfattah et al., 2021). Jacobian Covariance (Jacov) assesses the linear independence of feature maps. Let $\mathbf{J}(\mathbf{x}_i) = \nabla_{\mathbf{x}} f(\mathbf{x}_i; \theta)$ be the Jacobian vector of the output with respect to input $\mathbf{x}_i$. We compute the correlation matrix

$\mathbf{C} \in \mathbb{R}^{N \times N}$, where $\mathbf{C}_{ij}$ is the Pearson correlation coefficient between $\mathbf{J}(\mathbf{x}_i)$ and $\mathbf{J}(\mathbf{x}_j)$. The score is formulated as:

$$S_{jacov} = -\log \det(\mathbf{C} + \epsilon \mathbf{I}) \tag{18}$$

where $\mathbf{I}$ is the identity matrix and $\epsilon = 10^{-5}$ ensures numerical stability.

**3. SynFlow** (Abdelfattah et al., 2021). Synaptic Flow measures the capacity of the network to propagate signals without layer collapse. Deviating from the random Gaussian input, it utilizes a linearized objective $\mathcal{R} = \mathbb{K}^T f(\mathbf{x}_{ones}; \theta)$ computed on an **all-ones input tensor** $\mathbf{x}_{ones}$. The score is the sum of the "virtual" gradient-parameter product:

$$S_{synflow} = \sum_{\theta_k \in \theta} \left| \frac{\partial \mathcal{R}}{\partial \theta_k} \odot \theta_k \right| \tag{19}$$

where $\theta_k$ represents individual parameters and $\odot$ denotes the element-wise product.

**4. GraSP** (Abdelfattah et al., 2021). Gradient Signal Preservation (GraSP) captures the training dynamics by analyzing the Hessian-gradient product. Given the loss function $\mathcal{L}(\theta; \mathcal{B})$ computed on the synthetic batch (using the random labels $y$), let $\mathbf{g} = \nabla_\theta \mathcal{L}$ be the gradient and $\mathbf{H} = \nabla_\theta^2 \mathcal{L}$ be the Hessian matrix. The score is defined as:

$$S_{grasp} = -(\mathbf{H}\mathbf{g})^T \cdot \theta \tag{20}$$

In practice, this is computed efficiently via Hessian-vector products without explicit Hessian materialization. A higher score indicates better preservation of gradient flow after pruning.

**5. L2-Norm** (Abdelfattah et al., 2021). Benchmarked as a strong baseline for zero-cost NAS, the L2-Norm of parameters proxies the initialization magnitude and model complexity. It is defined as the Euclidean norm of all learnable weights:

$$S_{l2} = \|\theta\|_2 = \sqrt{\sum_{\theta_k \in \theta} \theta_k^2} \tag{21}$$

This metric reflects the signal variance potential at initialization.

**6. Params**. The parameter count serves as a direct measure of model capacity and memory footprint. While simple, it acts as a critical regularizer for the evaluator to balance performance against resource constraints:

$$S_{params} = \sum_{\theta_k \in \theta} \mathbb{I}(\theta_k \neq 0) \tag{22}$$

where $\mathbb{I}(\cdot)$ is the indicator function counting active learnable parameters.

**Appendix B.4. Initialization Details: Grid-Based Hybrid Dispersion Sampling**

To ensure the initial ground-truth evaluation set $\mathcal{P}_0$ achieves maximum geometric coverage across the search space, we implement the `SelectMaxDispersion` function using a Grid-Based Hybrid Sampling strategy. This approach synergizes the uniformity of MAP-Elites with the boundary-seeking capability of Farthest Point Sampling (FPS). The procedure consists of three steps:

**Step 1: Niche-Based Filtration.** First, we discretize the behavioral space into a grid of niches, identical to the structure of the Behavioral Atlas. We map all candidates from the zero-shot pool $\mathcal{P}_{pool}$ (synthesized by the agent) to their corresponding niches based on their behavioral coordinates. To enforce local uniformity, for each occupied niche, we retain only a single representative architecture that is geometrically closest to the niche's center:

$$p_{rep}^{(i)} = \arg\min_{p \in \text{Niche}_i} \|\Phi(p) - \mathbf{c}_i\|_2 \tag{23}$$

where $\Phi(p)$ denotes the behavioral coordinate of architecture $p$, and $\mathbf{c}_i$ is the centroid of the $i$-th niche. Let $\mathcal{S}_{rep}$ denote the set of these filtered representative architectures.

**Step 2: Cardinality-Conditioned Selection.** We then construct the final initial set $\mathcal{P}_0$ of target size $K$ based on the number of populated niches $|\mathcal{S}_{rep}|$:

- **Case A: Down-sampling** ($|\mathcal{S}_{rep}| > K$). If the number of populated niches exceeds the target budget $K$, we apply Farthest

Point Sampling (FPS) on the representative set $\mathcal{S}_{rep}$ to select exactly $K$ architectures. This ensures that we preserve the global boundary structure of the populated regions while discarding redundant internal points.

- **Case B: Up-sampling** ($|\mathcal{S}_{rep}| < K$). If the populated niches are insufficient to meet the budget, we first initialize $\mathcal{P}_0$ with all representatives: $\mathcal{P}_0 \leftarrow \mathcal{S}_{rep}$. To fulfill the remaining quota ($K - |\mathcal{S}_{rep}|$), we iteratively select candidates from the remaining unselected pool $\mathcal{R}$ (defined as $\mathcal{R} = \mathcal{P}_{pool} \setminus \mathcal{P}_0$) that maximize the distance to the *currently selected set*. In each iteration, the new architecture $p^*$ is chosen as:

$$p^* = \arg\max_{p \in \mathcal{R}} \left( \min_{s \in \mathcal{P}_0} \|\Phi(p) - \Phi(s)\|_2 \right) \tag{24}$$

This greedy mechanism aggressively explores the gaps between existing niches and potential outliers, ensuring the fixed budget $K$ is fully utilized to maximize dispersion.

### Appendix B.5. Detailed Formulation of Adaptive Strategy Scheduling

In Section 3.4 (Eq. 10), we introduced the Spatio-Temporal Gating Mechanism to regulate the exploration probability $P(\pi \in \Psi_{explore})$. This section details the precise functional definition of the temporal decay factor $\lambda(t)$ and the stagnation trigger $\beta_t$.

**1. Sigmoidal Temporal Decay Schedule ($\lambda(t)$).** Unlike linear decay schemes that apply constant pressure, we employ a non-linear **Sigmoidal transition function**. This schedule allows the agent to maintain a high exploration rate during the initial "accumulation phase" and rapidly transition to an exploitation-dominant regime after the mid-stage.

Let $\tau = t/T_{max}$ be the normalized search progress, where $t$ is the current generation and $T_{max}$ is the total budget. The decay function is formulated as:

$$\lambda(t) = \lambda_{start} - (\lambda_{start} - \lambda_{end}) \cdot \mathcal{S}(\tau) \tag{25}$$

where $\mathcal{S}(\tau)$ is the logistic sigmoid transition function defined as:

$$\mathcal{S}(\tau) = \frac{1}{1 + e^{-k \cdot (\tau - \tau_0)}} \tag{26}$$

**Hyperparameter Settings:**

- **Exploration Bounds:** $\lambda_{start} = 0.7$ and $\lambda_{end} = 0.25$. This ensures the exploration probability starts at 0.7 and smoothly decays to a strictly positive lower bound of 0.25 to prevent premature convergence.

- **Transition Steepness:** $k = 8.0$. A higher $k$ induces a sharper phase transition between exploration and exploitation.

- **Phase Midpoint:** $\tau_0 = 0.5$. The transition is centered at 50% of the total generations.

**2. Atlas Coverage Rate ($\Omega_t$).** The Behavioral Atlas ($\mathcal{M}_{state}$) partitions the latent behavioral space into $K$ distinct regions (niches) using Centroidal Voronoi Tessellation (CVT). The coverage rate $\Omega_t$ quantifies the global exploration progress at generation $t$ as the ratio of occupied niches:

$$\Omega_t(\mathcal{M}_{state}) = \frac{1}{K} \sum_{i=1}^{K} \mathbb{I}\left(|r_i^{(t)}| > 0\right) \tag{27}$$

where $r_i^{(t)}$ represents the $i$-th niche in the atlas accumulated up to generation $t$, and $|r_i^{(t)}|$ denotes the number of archived architectures within that niche. The indicator function $\mathbb{I}(\cdot)$ equals 1 if the niche is non-empty (occupied), and 0 otherwise. This metric serves as a state-aware proxy for the diversity saturation of the current population.

**3. Stagnation-Aware Dynamic Coefficient ($\beta_t$).** The coefficient $\beta_t$ serves as a reactive trigger to compensate for sparsity when the Behavioral Atlas stops growing. It is defined as a discrete switch:

$$\beta_t = \begin{cases} \beta_{base} & \text{if } \Delta\Omega_t < \epsilon \quad \text{(Stagnation Detected)} \\ 0 & \text{otherwise} \end{cases} \tag{28}$$

where $\Delta\Omega_t$ denotes the coverage gain over a sliding window (e.g., 5 generations), $\epsilon = 10^{-4}$ is the stagnation threshold, and $\beta_{base} = 0.2$ determines the magnitude of the re-exploration boost.

# Appendix C. Additional Experimental Results & Analysis

### Appendix C.1. Ablation Study Implementation Protocols

This section provides the exact implementation details for the ablation variants discussed in Section 4.2.1, clarifying the replacement strategies for removed components to ensure reproducibility.

**1. Variant: w/o Semantic Agent (Standard Random Evolution).**
In this variant, the Semantic Agent ($\mathcal{A}_\pi$) is removed to assess the necessity of high-level reasoning. Consequently, the architecture generation relies on standard stochastic operators rather than semantic planning. The operations are defined as follows:

- **Random Mutation:** We apply random perturbations to the operation sequence. Specifically, we randomly select $k$ positions (typically $k = 1$ or $2$) within the architecture and replace their operations with new ones sampled uniformly from the search space.

- **Random Crossover:** We employ a stochastic recombination strategy. Given two parent architectures, the offspring is constructed by randomly inheriting operations from either parent at each position. This process effectively mixes the functional components of the two parents without any semantic logic or validity checks.

**2. Variant: w/o Graph Evaluator (Random Selection).**
In the full SAGE-NAS framework, the Graph Evaluator ($\mathcal{E}_\phi$) acts as a surrogate model to rank the large pool of candidates generated by the agent, selecting only the top-$k$ for ground-truth evaluation. When this module is removed, we bypass the prediction-based filtering step. Instead, from the pool of offspring generated by the Semantic Agent, we employ a Random Selection strategy to choose the candidates for ground-truth evaluation, up to the maximum budget per generation. This ablation effectively tests the efficiency of the evaluator in identifying premium architectures from the generated candidates.

**3. Variant: w/o Behavioral Atlas (Fitness-Centric Elitism).**
When the Behavioral Atlas ($\mathcal{M}_{state}$) is removed, the *Sparsity-Driven Feedback Loop* and the *Spatio-Temporal Gating Mechanism* (Eq. 8) are deactivated. The system no longer maps architectures to the low-dimensional behavioral space utilizing Centroidal Voronoi Tessellation (CVT) niches, nor does it enforce the complementary *[Resource, Trainability, Expressivity]* diversity constraints. Consequently, the population management strategy reverts to a standard Fitness-Centric Elitism. In this setting, the scheduler maintains the population by strictly retaining the individuals with the highest ground-truth accuracy, effectively ignoring the geometric spatial dispersion of the candidates.

### Appendix C.2. Parameter Scaling of the LLM Backbone

To complement the cross-model benchmark in Section 4.2.2, we further investigate the impact of parameter scaling within the Qwen model family on the NAS-Bench-201 search space.

As visualized in Table 11, we observe a distinct Performance-Scale Correlation: search efficacy improves monotonically with the expansion of the LLM backbone. While smaller models (e.g., Qwen3-8b and 14b) are capable of generating valid candidates, their limited instruction-following capabilities often lead to suboptimal interpretation of the Behavioral Atlas's state feedback, resulting in higher performance variance (e.g., $\pm 0.42\%$ on CIFAR-100). In contrast, Qwen-plus benefits from the reasoning abilities associated with model scaling. It not only achieves the highest mean accuracy across all datasets but also exhibits exceptional convergence stability, attaining 0.00 variance on both CIFAR-10 and CIFAR-100. This specifically validates that SAGE-NAS effectively leverages the advanced cognitive capacity of large-scale models to precisely navigate the semantic space, ensuring the reliable identification of the global optimum (e.g.,94.37% on CIFAR-10).

### Appendix C.3. Visual Analysis of Behavioral Atlas and Elite Landscape

Figure 4 visualizes the evolutionary trajectory and final population distribution of SAGE-NAS within the DARTS search space. To construct a compact and discriminative behavioral atlas, we implement a standardized normalization for the metric axes: the Resource dimension ($\Phi_{param}$) is measured in Millions (M); the Trainability dimension ($\Phi_{syn}$) utilizes $\log_{10}$-transformed SynFlow values to mitigate magnitude disparities; and the Expressivity dimension ($\Phi_{nwot}$) adopts the NWOT metric to quantify the diversity of linear activation regions. Finally, the validation accuracy indicated by the color scale corresponds to the proxy performance obtained from the 12-epoch training protocol employed during the search phase.

*Table 11.* Impact of LLM parameter scaling on SAGE-NAS performance. We compare models of varying capacities within the Qwen family on the NAS-Bench-201 search space. The results illustrate a positive correlation between model cognitive capacity and search efficacy, with Qwen-plus achieving near-perfect convergence stability and global optimality.

| LLMs | NAS-Bench-201 (Test Acc %) | | |
| --- | --- | --- | --- |
| | CIFAR-10 | CIFAR-100 | ImageNet-16 |
| Qwen3-8b | $93.24 \pm 0.35$ | $71.99 \pm 0.37$ | $45.86 \pm 0.50$ |
| Qwen3-14b | $93.26 \pm 0.24$ | $72.03 \pm 0.42$ | $45.75 \pm 0.48$ |
| Qwen3-32b | $93.99 \pm 0.27$ | $72.13 \pm 0.32$ | $46.22 \pm 0.45$ |
| **Qwen-plus** | $\mathbf{94.37 \pm 0.00}$ | $\mathbf{73.51 \pm 0.00}$ | $\mathbf{47.05 \pm 0.31}$ |
| Optimal | 94.37 | 73.51 | 47.31 |

The visualization reveals the following critical topological characteristics:

- **CVT-based Atlas Discretization:** Consistent with the elite retention strategy employed by SAGE-NAS in the 3D space, the projection plane at the bottom illustrates this *high-dimensional discretization mechanism* as a marginalized landscape on the Resource-Trainability subspace. Each polygonal cell represents a topological niche, where the color maps the performance upper bound of the retained elite within that local 3D region via a "max-pooling" strategy. This visualization effectively delineates the global performance landscape, demonstrating how the algorithm precisely localizes high-performance regions while maintaining structural diversity across the behavioral space.

- **Sparsity-Driven Balance:** The broad spatial coverage of the 3D scattered points indicates a successful dynamic equilibrium between exploration and exploitation. The presence of architectures in sparse boundary regions evidences the effectiveness of the *Novelty-Seeking* mechanism in expanding search boundaries and preventing mode collapse, ensuring that the algorithm sufficiently explores the potential feasible regions within the behavioral space.

## Appendix C.4. Evolutionary Trajectory Analysis

To complement the static metrics presented in Table 5, Figure 5 visualizes the evolutionary trajectories of all variants throughout the search process. These curves offer a temporal perspective on how each component dictates convergence behavior:

- **Semantic Agent:** The dark orange curve (*w/o Semantic Agent*) exhibits the lowest starting point and a sluggish ascent across all tasks. The wide variance bands (shaded regions) indicate high instability, confirming that the Semantic Agent is crucial for addressing the "cold start" problem by providing high-quality initialization that stochastic mutation alone cannot match.

- **Behavioral Atlas:** The green curve (*w/o Behavioral Atlas*) displays a characteristic "greedy" pattern: rapid initial gains followed by premature stagnation (plateau). In contrast, SAGE-NAS (golden yellow curve) leverages sparsity-driven exploration to maintain population diversity. While this strategy incurs a slight exploration cost initially, it enables a significant "late-stage breakthrough" (e.g., around Gen 40 on CIFAR-10), effectively escaping local optima.

- **Graph Evaluator:** The blue curve (*w/o Graph Evaluator*) suffers from a slower convergence rate and a sub-optimal ceiling compared to the full model. This indicates that without the ranking fidelity provided by the Dual-Modality Graph Evaluator, the system struggles to precisely distinguish premium architectures from merely good ones, thereby limiting the final performance.

In summary, the full SAGE-NAS framework (golden yellow) integrates these modules to achieve not only superior accuracy but also the narrowest variance bands, demonstrating robust and stable search dynamics.

## Appendix C.5. Visualization of Discovered Architectures

In this section, we visualize the optimal architecture genotype discovered by SAGE-NAS on the ImageNet dataset within the standard DARTS search space. As shown in Figure 16, the searched architecture consists of two types of cells: the Normal Cell (Fig. 16a), which preserves the spatial resolution, and the Reduction Cell (Fig. 16b), which reduces the feature map size and increases the channel depth.

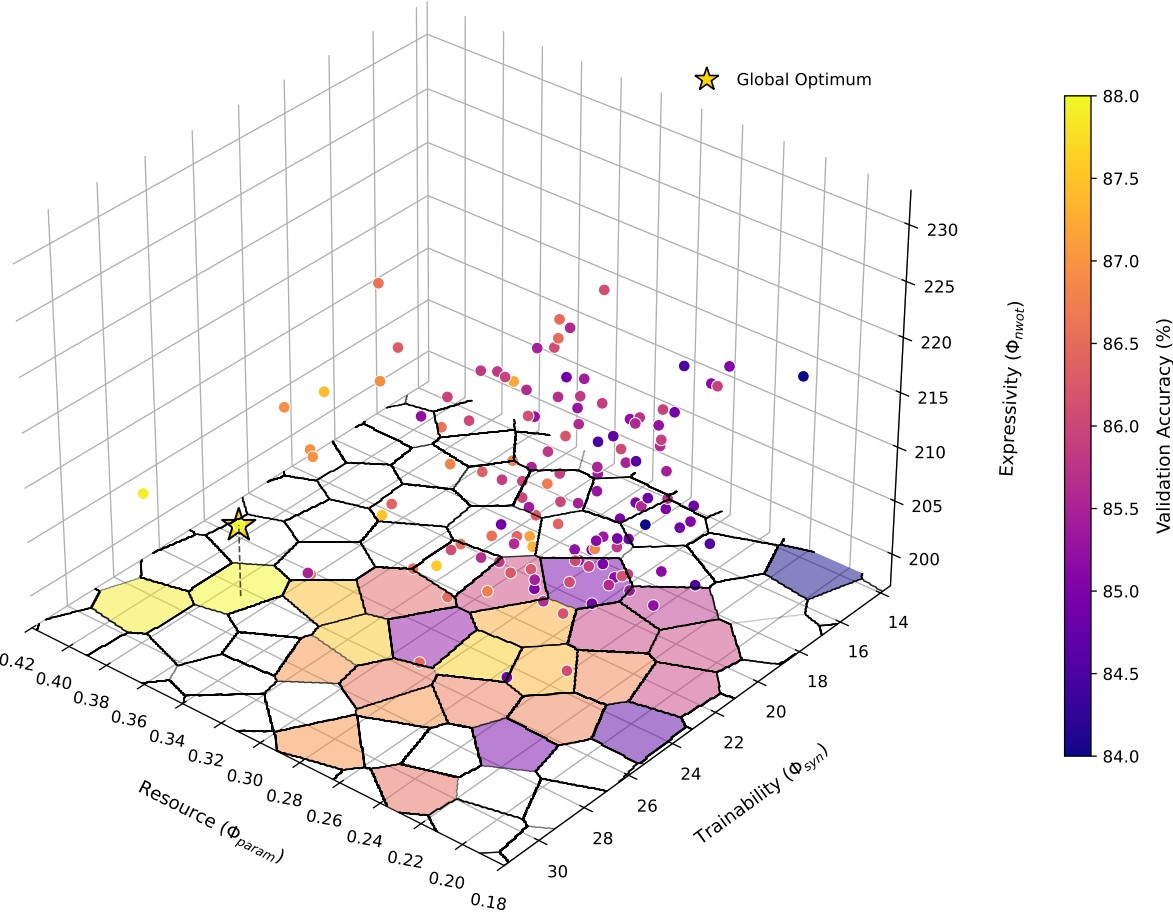

*Figure 4.* Visualization of the SAGE-NAS Behavioral Atlas on DARTS. The 3D scatter plot depicts the final population mapped onto the behavioral atlas defined by Params (M), $\log_{10}$(SynFlow), and NWOT. The bottom plane displays the *Projected Performance Landscape*, where the search space is discretized into Voronoi niches via CVT. The color of each niche corresponds to the validation accuracy of the local elite, providing a structured view of the search history.

### Appendix C.6. Analysis of the Dual-Modality Graph Evaluator

In this section, we provide an in-depth analysis of the Dual-Modality Graph Evaluator, focusing on its computational overhead, the efficacy of the pre-training tasks, and the anti-overfitting mechanisms of the HyperNetwork in few-shot scenarios. The quantitative results of our joint ablation study are summarized in Table 12.

**Computational Overhead and Pre-training Validation.** Regarding the additional computational cost introduced by the graph evaluator, we emphasize that the topological pre-training is strictly a *one-time overhead* per unique search space. Once the model learns the universal topological features via unsupervised learning, it seamlessly transfers to all downstream tasks within that space at zero additional cost. Furthermore, to validate the design of the pre-training tasks, we ablate the structural (*w/o Node Degree*) and semantic (*w/o Op. Reconstruction*) reconstruction objectives. As shown in Table 12, both mechanisms are indispensable for capturing robust topological priors; the absence of either leads to a degradation in representational capacity and subsequently limits the final search performance.

**Anti-Overfitting Mechanism of the HyperNetwork.** During the online evaluation phase of evolutionary search, the budget for ground-truth evaluations is extremely limited. In such data-scarce loops, traditional "static adapters" are highly prone to severe overfitting by memorizing the limited sample buffer. To address this challenge, the HyperNetwork in our framework is introduced not merely to expand model capacity, but to act as a powerful *structural regularizer*. By dynamically generating adapter weights conditioned on the input topology, the HyperNetwork forces the evaluator to learn a generalized meta-mapping, explicitly preventing the rote memorization of small sample sets. As evidenced in Table 12, this dynamic adaptation mechanism significantly outperforms the static adapter baseline in both accuracy and convergence

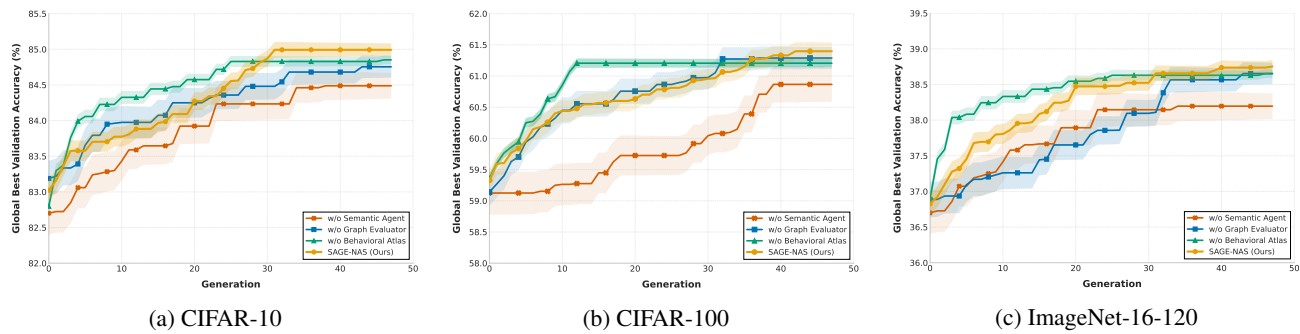

(a) CIFAR-10          (b) CIFAR-100          (c) ImageNet-16-120

*Figure 5.* Evolutionary trajectories of SAGE-NAS and ablation variants. The plots display the global best validation accuracy over generations on three datasets. Solid lines represent the mean accuracy, and shaded areas indicate the standard deviation across independent runs.

stability.

*Table 12.* Joint ablation study of the Dual-Modality Graph Evaluator on the NAS-Bench-201 search space. The results validate the necessity of the pre-training objectives and the superiority of the dynamic HyperNetwork over a static adapter. **Bold** values indicate the best performance.

| Ablation Variant | CIFAR-10 (%) | CIFAR-100 (%) | ImageNet-16 (%) |
|---|---|---|---|
| w/o Pre-training | $94.06 \pm 0.22$ | $73.21 \pm 0.21$ | $45.97 \pm 0.63$ |
| w/o Node Degree | $94.21 \pm 0.15$ | $73.33 \pm 0.18$ | $46.07 \pm 0.49$ |
| w/o Op. Reconstruction | $94.24 \pm 0.12$ | $73.30 \pm 0.21$ | $46.34 \pm 0.37$ |
| Static Adapter | $94.16 \pm 0.08$ | $73.27 \pm 0.22$ | $46.18 \pm 0.55$ |
| **Ours (Full Model)** | $\mathbf{94.37 \pm 0.00}$ | $\mathbf{73.51 \pm 0.00}$ | $\mathbf{47.05 \pm 0.31}$ |

## Appendix C.7. Further Justifications of Algorithmic Designs

To further validate the rationale behind the core algorithmic designs of SAGE-NAS, this section provides supplementary ablation studies focusing on the selection of behavioral atlas dimensions and the maximum dispersion strategy during population initialization.

**Selection and Complementarity of Behavioral Axes.** Within the State-Aware Behavioral Atlas, we construct a complementary "Expressivity-Trainability-Resource" 3D space grounded on the empirical findings of NAS-Bench-Suite-Zero (Krishnakumar et al., 2022). Specifically, NWOT (expressivity) and SynFlow (trainability) not only exhibit top-tier correlations with ground-truth accuracy but also demonstrate exceptionally strong complementarity. Furthermore, incorporating Params/FLOPs acts as a strict resource constraint, preventing the algorithm from "cheating" to artificially inflate Zero-Cost Proxy (ZCP) scores via infinite layer stacking. To empirically validate this configuration, we compare our chosen axes against low-complementarity baselines (e.g., SNIP + Jacov) on NAS-Bench-201. As shown in Table 13, our configuration achieves the highest atlas coverage (20.3%) and consistently yields the optimal search accuracy across all datasets, underscoring its superior discriminative capacity.

*Table 13.* Ablation study on the selection of Behavioral Axes within the NAS-Bench-201 search space. **Bold** values indicate the best performance and highest atlas coverage.

| Behavioral Axes | Atlas Coverage (%) | CIFAR-10 (%) | CIFAR-100 (%) | ImageNet-16 (%) |
|---|---|---|---|---|
| SNIP + Jacov + Params/FLOPs | $16.8 \pm 2.1$ | $93.67 \pm 0.39$ | $71.70 \pm 0.45$ | $45.87 \pm 0.38$ |
| GraSP + Jacov + Params/FLOPs | $19.1 \pm 1.6$ | $94.14 \pm 0.15$ | $73.09 \pm 0.32$ | $46.22 \pm 0.25$ |
| **Ours (NWOT + SynFlow + Params/FLOPs)** | $\mathbf{20.3 \pm 1.2}$ | $\mathbf{94.37 \pm 0.00}$ | $\mathbf{73.51 \pm 0.00}$ | $\mathbf{47.05 \pm 0.31}$ |

**Effectiveness of Maximum Dispersion in Initialization.** Although a maximum dispersion strategy might be suboptimal for unimodal optimization landscapes, *SelectMaxDispersion* in SAGE-NAS explicitly mitigates the "Semantic Collapse" problem inherent to LLM-driven NAS. Because pre-trained LLMs often exhibit a generative bias towards homogeneous

architectures in highly multimodal NAS spaces, we enforce maximum dispersion *strictly* during the initialization phase (Phase 1) to forcibly break this bias. Subsequently, the local exploitation phase ($\pi_{local}$) efficiently redirects the computational budget to high-density elite regions. The ablation results presented in Table 14 confirm that omitting this dispersion control leads to premature convergence and severely degrades cross-task generalization, particularly on more complex datasets.

*Table 14.* Ablation study on the initialization strategy within the NAS-Bench-201 search space. **Bold** values indicate the best performance.

| Method | CIFAR-10 (%) | CIFAR-100 (%) | ImageNet-16 (%) |
|---|---|---|---|
| w/o SelectMaxDispersion | $94.36 \pm 0.00$ | $73.17 \pm 0.08$ | $46.48 \pm 0.00$ |
| **SAGE-NAS (Ours)** | $\mathbf{94.37 \pm 0.00}$ | $\mathbf{73.51 \pm 0.00}$ | $\mathbf{47.05 \pm 0.31}$ |

## Appendix D. Prompt Engineering, Symbolic Representation & Reasoning Protocols

In this section, we detail the core interaction mechanisms driving the Semantic Agent ($\mathcal{A}_\pi$) and present the specific prompt templates used to reproduce our experiments. To reconcile computational efficiency with search precision, we established a High-Throughput Structured Reasoning Protocol.

Unlike conversational agents that rely on high-latency multi-turn interactions, SAGE-NAS pioneers a Single-Step Generation Paradigm. As illustrated in Figures 6–9, this protocol compresses the complex evolutionary process into a strict "Input Mapping $\rightarrow$ Symbolic Decoding" workflow. In each evolutionary iteration, the system encapsulates the Strategy-Guided System Instruction and the Context Data (containing current architecture state and hard constraints) into a complete Token Stream. This One-pass Input design not only eliminates the inference latency associated with multi-turn dialogues but also significantly enhances the iteration speed of population evolution and the stability of instruction following via the joint attention mechanism of the context window.

### Appendix D.1. Instruction Design Paradigms for Evolutionary Strategies

We translate the abstract evolutionary operators defined in Section 3.2 into logic instructions executable by Large Language Models (LLMs). Through Cognitive Decomposition, these prompts trigger specific reasoning paths within a single forward pass:

- **Exploitation Instruction ($\pi_{local}$): Macroscopic Freezing & Microscopic Refinement.** Aligned with the constrained symbolic editing defined in Eq. (2), the prompt (Fig. 6) imposes a hard constraint of "Topological Freezing." It mandates the model to maintain the macroscopic DAG skeleton while identifying and replacing semantically similar operators, thereby mining local extrema to ensure the offspring inherits superior structural genes while pushing towards the local optimum.

- **Exploration Instruction ($\pi_{global}$): Semantic-Driven Macroscopic Reconstruction.** To break through performance bottlenecks caused by local stagnation, the prompt (Fig. 7) establishes a "Semantic Intervention" mechanism. Instead of optimizing a simple distance metric, the input instruction explicitly directs the model to disrupt existing connection patterns and alter connectivity states (e.g., pruning or activation). This drives the agent to execute a "semantic jump," effectively escaping local attractors to explore sparse fringes potentially harboring superior optima.

- **Crossover Instruction ($\pi_{cross}$): Chain-of-Thought Synergistic Synthesis.** To address the structural breakage issue, we reframe the crossover operator as a Chain-of-Thought (CoT) synthesis process. The prompt (Fig. 8) guides the model to perform three-stage reasoning: Analyzing complementary strengths, Planning a synergistic fusion blueprint, and Decoding it into a valid topology. This ensures the generated offspring logically transcends the performance limits of its parents rather than being a random hybrid.

- **Novelty-Seeking Instruction ($\pi_{zero}$): Principle-Driven Ab Initio Construction.** Targeting sparse regions (OOD) in the Behavioral Atlas, the prompt (Fig. 9) activates the Zero-Shot Generation mode. The instruction severs dependence on the current population, relying instead on abstract Design Principles to build unique topological structures from scratch (*Ab Initio*). This mechanism injects high-potential "design genes" into the population to efficiently prevent mode collapse while maintaining intrinsic trainability.

**Appendix D.2. Symbolic Specification and Prior Injection of the Search Space**

The execution of the evolutionary strategies described above relies on the Semantic Agent's precise adherence to the physical boundaries of the search space. To achieve this, we implement a Modular Prompt Injection Mechanism.

Specifically, the placeholder {constraints} appearing in the Context & Task block of the strategy protocols (Figures 6–9) is dynamically instantiated at runtime with space-specific logic. We have designed four distinct Constraint Modules corresponding to mainstream NAS benchmarks. These modules are not merely format descriptions but are injected as Immutable Axioms, ensuring the LLM constructs a valid "Topological Cognitive Map" before generating any architecture. The correspondence between the search spaces and their symbolic constraint prompts is detailed below:

1. **DARTS Module (Fig. 10): Axiomatic Enforcement of DAG Causality.** Targeting the unrestricted topology of DARTS, this module instantiates the {constraints} slot with strict mathematical inequalities ($src < current\_node$). By converting the abstract topological property of "acyclicity" into explicit Symbolic Logic Predicates, we compel the LLM to perform strictly causal reasoning. This ensures that every generated Genotype guarantees unidirectional data flow, effectively eliminating the risk of cyclic (invalid) graphs in deep stacking.

2. **NAS-Bench-201 Module (Fig. 11): Semantic Decoding of Fixed-Skeleton Logic.** For the string-based NB201 space, the injected constraint goes beyond mere syntax checking to clarify the underlying "Summation Aggregation" mechanism. By explicitly describing the feature map aggregation logic ($N_0 \rightarrow N_3$) within the prompt, we enable the LLM to comprehend the physical impact of operator selection. This promotes an understanding of the computation graph's connectivity implicit in the string representation, rather than treating it as a simple text generation task.

3. **NAS-Bench-101 Module (Fig. 12): Budget-Aware Combinatorial Planning.** As depicted in Fig. 12, this module transforms the rigid constraint of "Total Edges $\leq 9$" into a Resource Budgeting Problem. Instead of a passive numerical limit, the prompt frames edge allocation as a reasoning task, requiring the LLM to dynamically balance network depth and width. This acts as a mathematical regularizer, guiding the agent to construct topologically efficient structures within the strict allowable budget.

4. **TransNAS-Bench-101 Module (Fig. 13): Semantic Reinforcement of Topological Connectivity.** Addressing the sparse connectivity risks in this lightweight space, the constraint module specifically redefines the zero operator as a "Physical Disconnection" rather than a neutral label. We introduce explicit validity axioms (e.g., "non-all-zero input checks") that serve as Differentiability Guardrails. This guides the LLM to maintain the integrity of the gradient flow path even during aggressive sparsity-driven exploration.

**Appendix D.3. Mechanism of Geometric-Semantic Translation**

To bridge the modality gap between the continuous Behavioral Atlas ($\mathcal{M}_{state}$) described in Section 3.4 and the discrete symbolic reasoning required by the Semantic Agent ($\mathcal{A}_{\pi}$), SAGE-NAS implements a Geometric-Semantic Translator. This module functions as the system's "cognitive bridge," dynamically synthesizing the feedback payloads that are subsequently injected into the {feedback} placeholder of the evolutionary strategy prompts (as detailed in Figure 6 and Figure 9).

**Visual Grounding and Integrated Data Flow.** Figure 14 visualizes the internal logic of this translation process. Unlike standard prompting which might separate data and instruction, our mechanism constructs a unified composite payload injected entirely into the {feedback} slot. This payload consists of two synchronized components derived from the input:

- **Contextual Grounding (Derived from Left Path):** The raw architecture information (e.g., coordinates and JSON structures of Ref_A, Elite_1 shown in the *Input* blocks of Fig. 14) is embedded directly within the feedback message. This provides the LLM with the concrete *objects* of manipulation (the "What").

- **Semantic Directive (Derived from Right Path):** Simultaneously, the Translator analyzes these inputs to synthesize high-level geometric reasoning (shown in the *Output* blocks of Fig. 14). These directives—ranging from geometric gap analysis to induced structural priors—provide the specific *instructions* for manipulation (the "How").

By consolidating both the raw topological context and the semantic guidance into a single {feedback} injection, the system ensures that the LLM receives a self-contained, physics-aware instruction block that tightly couples geometric goals with architectural constraints.

**Translation Paradigms.** As illustrated in Figure 14, the translator adapts its logic based on the geometric state of the target Behavioral Atlas region:

- **Case 1: Differential Navigation for Exploration ($\pi_{zero}$).** When targeting sparse regions, the system lacks direct templates. As visualized in the "Gap Analysis" block of Figure 14 (Case 1), the Translator executes a rule-based differential projection. Specifically, it calculates numerical deviations between the geometric boundaries of the targeted unexplored regions and the nearest neighbor anchors to deterministically map state differences to semantic descriptors (e.g., $\Delta P < 0 \rightarrow$ "Insufficient Capacity"). It then synthesizes a navigational feedback block that instructs the agent to interpolate between known landmarks, explicitly defining the required shifts in Resource, Trainability, and Expressivity to populate the void.

- **Case 2: Atlas-Guided Induction for Exploitation ($\pi_{local}$).** For high-performing elite clusters, the system implements a Chain-of-Induction mechanism. Instead of relying on hand-crafted rules, the Translator invokes an inductive reasoning step using the LLM backbone. This process is governed by a specialized Design Principle Extraction Protocol (visualized in Figure 15), which explicitly instructs the LLM to dynamically distill common topological patterns from the Top-5 elites into abstract "Design Principles" (as shown in Step 1 of Fig. 14). These distilled principles are then integrated with the region's Physical Constraints (Step 2) to form a holistic refinement advisory. This ensures that the generated offspring inherit the superior structural traits of the local cluster while strictly adhering to the geometric boundaries of the target Elite Regions.

**Appendix D.4. Robustness Protocol: Handling Generation Failures**

Given the stochastic nature of Large Language Models, generating strictly valid symbolic structures (e.g., DAGs with specific edge constraints) presents inherent challenges. To ensure the stability of the evolutionary process, SAGE-NAS implements a Three-Tier Validation and Repair Mechanism. This protocol activates immediately after the symbolic decoding step and before the architecture is admitted to the population.

**1. Tier-1: Syntax Validation (JSON Parsing).** The raw text stream generated by the Agent is first passed through a strict JSON parser.

- *Failure Mode:* Malformed JSON (e.g., unclosed brackets, missing delimiters) caused by token truncation or format hallucination.
- *Handling Strategy:* These format errors are treated as transient generation failures. The system immediately triggers an Automatic Retry ($N_{retry} = 3$). If the output remains unparsable after all attempts, the mutation operation is aborted.

**2. Tier-2: Schema Verification.** Successfully parsed JSON objects are validated against the strict schema definitions provided in Section Appendix D.2.

- *Failure Mode:* Missing keys (e.g., omitting `normal_concat` in DARTS), type mismatches, or referencing operations outside the allowed primitives ($\mathcal{O}$).
- *Handling Strategy:* These instances are treated as "Instruction Following Failures." The system rejects the sample and decrements the retry counter, triggering a regeneration with the same prompt.

**3. Tier-3: Topological Feasibility Check (Constraint Enforcement).** This final stage verifies the "Immutable Axioms" injected via the constraints (Figs. 10–13). Instead of blindly discarding computationally expensive samples that satisfy syntax but violate complex logic, we apply deterministic Heuristic Repair:

- *Cycle Breaking (DARTS):* If a causal violation ($src \geq node_{idx}$) is detected, the offending edge is pruned or re-routed to the nearest valid predecessor ($src = 0$).
- *Budget Truncation (NAS-Bench-101):* If the edge count exceeds the budget ($|E| > 9$), edges with the lowest weight (if weighted) or random edges are pruned until the constraint is met.
- *Connectivity Restoration (TransNAS-Bench-101):* If a `zero` operator causes a path disconnection, it is replaced with a minimal-cost primitive (e.g., `skip_connect`) to restore gradient flow.

---

**Exploitation Strategy Protocol ($\pi_{local}$)**

⌨ **Input Prompt**

🖥 **System Instruction:**

You are an expert in Fine-grained Architecture Refinement. The current strategy is Homology-Preserving Refinement.
**Goal:** Mine local extrema around the provided elite architecture.
**Constraint:** You must FREEZE the macroscopic topology. Focus solely on optimizing microscopic operators.

---

🧠 **Context & Task:**

**Input Elite ($\alpha^*$):** Score = $\{$`eval_score:.4f`$\}$, $\{$`architecture_json`$\}$
**Task:** Perform a conservative mutation to approximate the local optimum.
**Constraints:** $\{$`constraints`$\} \leftarrow$ Hard boundaries (e.g., DARTS/NB201 spaces).
**Feedback:** $\{$`feedback`$\} \leftarrow$ Insights from Behavioral Atlas.
**Actionable Rules:**

- **Structure:** Keep the DAG skeleton exactly the same. Do not add/remove edges.
- **Operation:** Identify 1 or 2 operations to swap. Prefer semantic neighbors.

**Output:** Return the refined architecture in strict JSON.

⇓ *LLM Inference* ⇓

</> **Semantic Agent Output**

```json
{
  "normal": [
    ('dil_conv_5x5', 0), ('skip_connect', 0),
    ('max_pool_3x3', 1), ('dil_conv_3x3', 0),
    ('avg_pool_3x3', 2), ('sep_conv_3x3', 3),
    ('skip_connect', 1), ('dil_conv_5x5', 2)
  ],
  "normal_concat": [2, 3, 4, 5],
  "reduce": [
    ('max_pool_3x3', 0), ('dil_conv_3x3', 1),
    ('avg_pool_3x3', 0), ('skip_connect', 0),
    ('max_pool_3x3', 1), ('avg_pool_3x3', 3),
    ('dil_conv_5x5', 0), ('skip_connect', 2)
  ],
  "reduce_concat": [2, 3, 4, 5]
}
```

*Figure 6.* Visualization of the Exploitation Strategy Protocol ($\pi_{local}$). This mechanism facilitates Homology-Preserving Refinement. The prompt instructs the agent to strictly freeze the macroscopic topology while performing fine-grained mutations on microscopic operators to mine local extrema, ensuring the offspring inherits the superior topological structure while pushing towards the local optimum.

## Exploration Strategy Protocol ($\pi_{global}$)

### ⌨ Input Prompt

**🖥 System Instruction:**

You are an expert in Macroscopic Topological Reconstruction.
**Goal:** Break away from local optima and performance bottlenecks by introducing significant structural diversity.
**Directive:** You must ALTER the macroscopic topology. Aim to maximize **structural dissimilarity** (i.e., generate a graph with a different connectivity pattern compared to the parent).

**🧠 Context & Task:**

**Input Parent ($\alpha^*$):** Score = {`eval_score:.4f`}, {`architecture`}
**Task:** Perform an aggressive topological mutation on this specific parent.
**Constraints:** {`constraints`} ← Hard boundaries of the search space.
**Actionable Rules:**

- **Structure (Aggressive):** Even if the parent performs well, you must disrupt its connection patterns. Actively add, remove, or re-route edges (or alter connectivity states like Zeroize/Skip).
- **Goal (Structural Divergence):** The offspring must be structurally distinct to explore regions with higher performance potential. Ensure information flow paths are fundamentally altered.

**Output:** Return the novel architecture in strict JSON

⇓ *LLM Inference* ⇓

### ⟨⟩ Semantic Agent Output

```
{
  "normal": [
    ('sep_conv_3x3', 0), ('sep_conv_5x5', 1),
    ('sep_conv_5x5', 0), ('sep_conv_3x3', 2),
    ('dil_conv_3x3', 1), ('skip_connect', 0),
    ('skip_connect', 1), ('sep_conv_5x5', 1)
  ],
  "normal_concat": [2, 3, 4, 5],
  "reduce": [
    ('sep_conv_3x3', 0), ('max_pool_3x3', 1),
    ('dil_conv_5x5', 2), ('sep_conv_3x3', 1),
    ('sep_conv_5x5', 2), ('skip_connect', 0),
    ('avg_pool_3x3', 1), ('skip_connect', 0)
  ],
  "reduce_concat": [2, 3, 4, 5]
}
```

*Figure 7.* Visualization of the Exploration Strategy Protocol ($\pi_{global}$). This mechanism executes Semantic-Driven Macroscopic Topological Reconstruction. In contrast to exploitation, the prompt explicitly directs the agent to aggressively alter connection patterns to maximize structural divergence, thereby driving the search process to break through performance bottlenecks and expand into sparse regions potentially harboring superior optima.

## Crossover Strategy Protocol ($\pi_{cross}$)

### ⌨ Input Prompt

**🖥 System Instruction:**

You are an expert in Hybrid Architecture Design.
**Goal:** Utilize Chain-of-Thought (CoT) reasonin to synthesize a superior offspring that transcends the performance limits of both parents.
**Directive:** Do NOT perform random splicing (which causes breakage). You must execute a three-stage synthesis: Analyze strengths → Plan fusion logic → Generate valid topology.

### 🧠 Context & Task:

**Input Parent A:** Score = $\{$`score_A:.4f`$\}$, $\{$`architecture_A`$\}$
**Input Parent B:** Score = $\{$`score_B:.4f`$\}$, $\{$`architecture_B`$\}$
**Task:** Fuse the complementary strengths of A and B to design a superior architecture.
**Constraints:** $\{$`constraints`$\} \leftarrow$ Hard boundaries of the search space.
**Actionable Rules (CoT Steps):**

1. **Analyze Strengths ($\varphi$):** Deconstruct Parent A and B to identify **complementary advantages** (e.g., A's efficient stem vs. B's dense tail).
2. **Logical Planning ($S$):** Construct asynergistic fusion plan to strategically combine these features for mutual enhancement.
3. **Valid Generation ($D$):** Instantiate this blueprint into a grammatically valid JSON object.

**Output:** Return the final synthesized JSON only.

⇓ *LLM Inference* ⇓

### ⟨⟩ Semantic Agent Output

```
{
 "normal": [
   ('sep_conv_3x3', 0), ('sep_conv_3x3', 1),
   ('sep_conv_3x3', 0), ('sep_conv_3x3', 1),
   ('sep_conv_5x5', 0), ('dil_conv_3x3', 2),
   ('skip_connect', 0), ('sep_conv_5x5', 1)
 ],
 "normal_concat": [2, 3, 4, 5],
 "reduce": [
   ('max_pool_3x3', 0), ('sep_conv_3x3', 1),
   ('sep_conv_5x5', 0), ('max_pool_3x3', 1),
   ('max_pool_3x3', 0), ('sep_conv_5x5', 2),
   ('skip_connect', 0), ('sep_conv_5x5', 1)
 ],
 "reduce_concat": [2, 3, 4, 5]
}
```

*Figure 8.* Visualization of the Crossover Strategy Protocol ($\pi_{cross}$). The prompt guides the agent to perform Chain-of-Thought (CoT) synthesis. Instead of random crossover, the agent explicitly analyzes the complementary strengths of two parents, logically plans their combination, and generates a valid architecture that synergistically surpasses the performance of both parents.

## Novelty-Seeking Strategy Protocol ($\pi_{zero}$)

### ⌨ Input Prompt

**🖥 System Instruction:**

You are a top expert proficient in Neural Architecture Search (NAS) and Evolutionary Algorithms. You are executing the Novelty-Seeking Task. Your goal is to populate unmapped regions of the Behavioral Atlas with high-potential candidates.

**Directive:** You must completely detach from existing patterns. Do NOT imitate the provided references. Instead, execute Zero-Shot Generation to build a brand-new architecture from scratch based strictly on abstract Design Principles to ensure intrinsic trainability.

**🧠 Context & Task:**

**Input Principles:** {design_principles} ← Abstract axioms (e.g., Asymmetric Aggregation).
**Constraints:** {constraints} ← Hard boundaries of the search space.
**Feedback:** {feedback} ← Insights from Behavior Atlas.
**Task:** Generate a unique and viable architecture via the Input Strategy:

1. **Principle Analysis:** Interpret 2-3 core principles for the current context.
2. **Behavioral Space Exploration:** Identify structural voids (e.g., unexplored op-combinations, counter-intuitive connections).
3. **Ab Initio Construction:** Start with a blank canvas. Build a unique topology to fill sparse regions, prioritizing structural uniqueness while adhering to axiomatic quality.

**Output:** Return the fresh architecture in strict JSON.

⇓ *LLM Inference* ⇓

### ⟨⁄⟩ Semantic Agent Output

```
{
  "normal": [
    ('sep_conv_3x3', 0), ('skip_connect', 1),
    ('sep_conv_3x3', 0), ('sep_conv_3x3', 1),
    ('sep_conv_5x5', 0), ('dil_conv_5x5', 2),
    ('sep_conv_5x5', 3), ('sep_conv_5x5', 1)
  ],
  "normal_concat": [2, 3, 4, 5],
  "reduce": [
    ('max_pool_3x3', 0), ('sep_conv_3x3', 1),
    ('sep_conv_5x5', 0), ('dil_conv_3x3', 1),
    ('max_pool_3x3', 0), ('sep_conv_5x5', 2),
    ('skip_connect', 0), ('dil_conv_3x3', 1)
  ],
  "reduce_concat": [2, 3, 4, 5]
}
```

*Figure 9.* Visualization of the Novelty-Seeking Strategy Protocol ($\pi_{zero}$). The mechanism switches to a Zero-Shot Generation mode to fill sparse regions in the behavioral atlas. The prompt instructs the agent to analyze abstract design principles, identify structural voids, and construct architectures *ab initio* (from scratch), explicitly avoiding known elite patterns to inject high-potential design genes.

## 📄 DARTS Search Space Constraints Prompt

**0. Cell Internal Representation Format (Schema)**

The architecture must follow this strict `Genotype` format. It defines a DAG with two cells (Normal/Reduce), each containing 4 intermediate nodes (Nodes 2-5).

```
Genotype(
    # Normal Cell: 8 edges (2 per node) for Nodes 2, 3, 4, 5
    normal=[
        (op1, src1), (op2, src2),   # Node 2 inputs (src in 0,1)
        (op3, src3), (op4, src4),   # Node 3 inputs (src in 0,1,2)
        (op5, src5), (op6, src6),   # Node 4 inputs (src in 0,1,2,3)
        (op7, src7), (op8, src8)    # Node 5 inputs (src in 0,1,2,3,4)
    ],
    normal_concat=[2, 3, 4, 5],     # Always concat all intermediate nodes

    # Reduce Cell: Same structure, handles spatial downsampling
    reduce=[
        (op9, src9),    (op10, src10),
        (op11, src11), (op12, src12),
        (op13, src13), (op14, src14),
        (op15, src15), (op16, src16)
    ],
    reduce_concat=[2, 3, 4, 5]
)
```

**1. Structural Prerequisites**

- **Dual-Cell Design:** Architecture consists of `normal` and `reduce` cells.
- **Node Definition:** Nodes 0 and 1 are inputs from previous layers. Nodes 2, 3, 4, 5 are intermediate computation nodes.
- **Edge Count:** Each intermediate node receives exactly **2 incoming edges**. Total edges = 4 nodes × 2 edges × 2 cells = 16 edges.

**2. Allowed Primitive Operations ($\mathcal{O}$)**

Only the following 7 primitives are permitted for `op`:

```
sep_conv_3x3,   sep_conv_5x5,   dil_conv_3x3,   dil_conv_5x5,   max_pool_3x3,   avg_pool_3x3,
skip_connect
```

**3. Strict Topological Constraints (Validation Rules)**

*Warning: Violation of these rules results in an invalid DAG.*

- **Causal Constraint:** For any edge, the source node index (`src`) must be strictly smaller than the current node index.
- **Index Bounds:**
    - Node 2 inputs: $src \in \{0, 1\}$
    - Node 3 inputs: $src \in \{0, 1, 2\}$
    - Node 4 inputs: $src \in \{0, 1, 2, 3\}$
    - Node 5 inputs: $src \in \{0, 1, 2, 3, 4\}$
- **Tuple Indexing Rule:** For the $i$-th tuple in the list ($0 \le i < 8$), the valid range for `src` is:

$$0 \le src < 2 + \lfloor i/2 \rfloor$$

**4. Macro-Architecture Context**

The network is built by stacking these cells: `Stem` → N × (`Normal`/`Reduce` **Pattern**) → `GlobalPool` → `Classifier`. Reduction cells are placed at $1/3$ and $2/3$ network depth.

*Figure 10.* The Constraint Module for DARTS. This content is dynamically injected into the `constraints` placeholder of the strategy protocols when searching on DARTS.

---

### 📑 NAS-Bench-201 Search Space Constraints Prompt

**0. Cell Internal Representation Format (Schema)**

The architecture is defined by a fixed string format representing a DAG with 4 nodes ($N_0$ to $N_3$). $N_0$ is the input. The string uses '—' to delimit edges and '+' to delimit nodes.

```
# Format: |Node1_Edges|+|Node2_Edges|+|Node3_Edges|
# Computation Logic: Node_j = Sum( op(Node_i) for connected i )

# Specific Example String:
arch_str = '|op1~0|+|op2~0|op3~1|+|op4~0|op5~1|op6~2|'

# Interpretation:
#   Node 1 = op1(Node 0)
#   Node 2 = op2(Node 0) + op3(Node 1)    <-- Summation Aggregation
#   Node 3 = op4(Node 0) + op5(Node 1) + op6(Node 2)
```

**1. Structural Prerequisites**

- **Topology:** 4 nodes (Node 0 is Input; 1, 2, 3 are Intermediate).
- **Total Edges:** Fixed at 6 edges total ($1 + 2 + 3$).
- **Standard Format:** `|op1~0|+|op2~0|op3~1|+|op4~0|op5~1|op6~2|`

**2. Allowed Primitive Operations ($\mathcal{O}$)**

Only the following 5 primitives are permitted for `op`:

`none, skip_connect, nor_conv_1x1, nor_conv_3x3, avg_pool_3x3`

**3. Core Topological Constraints & Strategy**

*Warning: Violation of edge rules results in an invalid graph.*
- **One-Hot Selection:** For each of the 6 fixed edges, select exactly **one** operation from $\mathcal{O}$.
- **Recommended Strategy (Conflict-Free):**
    - Part 1: Define 1 op for edge $0 \rightarrow 1$.
    - Part 2: Define 2 ops for edges $0 \rightarrow 2$ and $1 \rightarrow 2$.
    - Part 3: Define 3 ops for edges $0 \rightarrow 3$, $1 \rightarrow 3$, and $2 \rightarrow 3$.

**4. Macro-Architecture Stacking**

The full network consists of **15 Search Cells** stacked in 3 stages:
- **Stage 1:** $5 \times$ Cells ($C = 16$, $32 \times 32$) $\rightarrow$ Reduction ($C \rightarrow 2C$, Downsample)
- **Stage 2:** $5 \times$ Cells ($C = 32$, $16 \times 16$) $\rightarrow$ Reduction ($2C \rightarrow 4C$, Downsample)
- **Stage 3:** $5 \times$ Cells ($C = 64$, $8 \times 8$) $\rightarrow$ Global Pool $\rightarrow$ Classifier

**5. Quick Validation Checklist**

✓ Are all ops in the allowed list?
✓ Does the format strictly follow `|X|+|XX|+|XXX|`?
✓ **Critical:** No duplicate edges from the same source to the same target?

---

*Figure 11.* The Constraint Module for NAS-Bench-201. This symbolic block instantiates the `constraints` variable for fixed-skeleton search.

## 📄 NAS-Bench-101 Search Space Constraints Prompt

**0. Architecture Representation Format (Schema)**
The architecture is defined by `NB101Genotype`, consisting of a list of intermediate nodes (operation + input indices) and a list of output connections.

```
# Format: intermediate_nodes=[(op, [inputs])...], output_connections=[...]
NB101Genotype(
    intermediate_nodes=[
        ('conv1x1-bn-relu', [0]),      # Node 1: connects to Input(0)
        ('conv3x3-bn-relu', [0, 1]),   # Node 2: connects to 0 & 1
        ('maxpool3x3',      [1]),      # Node 3: connects to 1
        ('conv3x3-bn-relu', [2, 3]),   # Node 4: connects to 2 & 3
        ('conv1x1-bn-relu', [4])       # Node 5: connects to 4
    ],
    output_connections=[4, 5]          # Output connects to Node 4 & 5
)
```

**1. Structural Prerequisites**

- **Node Set:** 7 Nodes Total. Node 0 (Input), Nodes 1-5 (Intermediate), Node 6 (Output).
- **DAG Flow:** $Input \rightarrow Intermediate \rightarrow Output$.
- **Causal Rule:** Node $i$ can only receive input from Node $j$ where $j < i$.

**2. Allowed Primitive Operations ($\mathcal{O}$)**
Only the following 3 primitives are permitted for `op`:

`conv1x1-bn-relu, conv3x3-bn-relu, maxpool3x3`

**3. Strict Topological Constraints**
*Warning: The edge count limit is the most critical constraint.*
- **Max Edge Limit:** Total edges must be $\leq$ **9**.
- **Connectivity:** Every intermediate node must have at least one input. The output node must have at least one connection.

**4. Edge Budget Calculation Logic**
The total edge count is the sum of all input lists plus the output list length:

$$\text{Total Edges} = \sum_{i=1}^{5} \text{len}(\text{node}_i.\text{inputs}) + \text{len}(\text{output\_connections}) \leq 9$$

*Example from Section 0:* 1 (n1) + 2 (n2) + 1 (n3) + 2 (n4) + 1 (n5) + 2 (out) = 9

**5. Recommended Generation Strategy**

1. **Plan Budget:** Allocate the 9 edges before assigning ops.
2. **Assign Inputs:** Ensure $j < i$ for all connections.
3. **Connect Output:** Don't leave the graph dead-ended.
4. **Diversify Ops:** Mix convolution and pooling layers.

*Figure 12.* The Constraint Module for NAS-Bench-101. This module serves as the instantiation logic for the `constraints` slot.

### 📄 TransNAS-Bench-101 Micro Search Space Constraints Prompt

**0. Cell Internal Representation Format (Schema)**

The architecture follows the NAS-Bench-201 string format representing a 4-node DAG ($N_0 \rightarrow N_3$). The string uses '—' to delimit edges and '+' to delimit nodes.

```
# Format: |Node1_Edges|+|Node2_Edges|+|Node3_Edges|
# DAG Structure: 4 Nodes, 6 Edges. N0 is Input, N3 is Output.
# Computation: Node_j = Sum( op(Node_i) for connected i )

# Specific Example String:
arch_str = '|op1~0|+|op2~0|op3~1|+|op4~0|op5~1|op6~2|'

# Interpretation:
#   Node 1 = op1(N0)
#   Node 2 = op2(N0) + op3(N1)
#   Node 3 = op4(N0) + op5(N1) + op6(N2)
```

**1. Structural Prerequisites**

- **Topology:** Fixed 4-node DAG. Nodes 1, 2, 3 compute features.
- **Edge Count:** Fixed at 6 edges total ($1 + 2 + 3$).
- **Aggregation:** Element-wise **Summation**.

**2. Allowed Primitive Operations ($\mathcal{O}$)**

Only the following 4 primitives are permitted for `op`:

```
identity, zero, conv3x3, conv1x1
```

**3. Source & Validity Constraints**

*Warning: Violation of these rules results in an invalid architecture.*

- **Source Constraints (Fixed Topology):**
    - Part 1 (Edge 1): Must come from Node ˜0.
    - Part 2 (Edges 2, 3): Must come from Node ˜0 or ˜1.
    - Part 3 (Edges 4, 5, 6): From Node ˜0, ˜1, or ˜2.
- **Validity Rules (Zero-Op Check):**
    - **Rule 1:** Operations on the **first 3 edges** (edges 1, 2, 3) cannot *all* be `zero`.
    - **Rule 2:** Operations on the **3rd, 5th, and 6th edges** cannot *all* be `zero`.

**4. Macro-Architecture Stacking**

The network consists of cells stacked in specific stages:

- **Input → Stem →**
- **R_Stage_1:** 2 Cells (Reduction: $C \times 2, H/2$) →
- **N_Stage_1:** 2 Cells (Normal: Keep Size) →
- **R_Stage_2:** 2 Cells (Reduction: $C \times 2, H/2$) →
- **GlobalAvgPool → Classifier**

**5. Quick Validation Checklist**

✓ Are only `identity`, `zero`, `conv3x3`, `conv1x1` used?
✓ Does Part 1 have 1 op, Part 2 have 2 ops, Part 3 have 3 ops?
✓ **Critical:** Do edges 1-3 contain at least one non-zero op?

*Figure 13.* The Constraint Module for TransNAS-Bench-101. Injected into the `constraints` field, this module acts as a gradient flow integrity constraint.

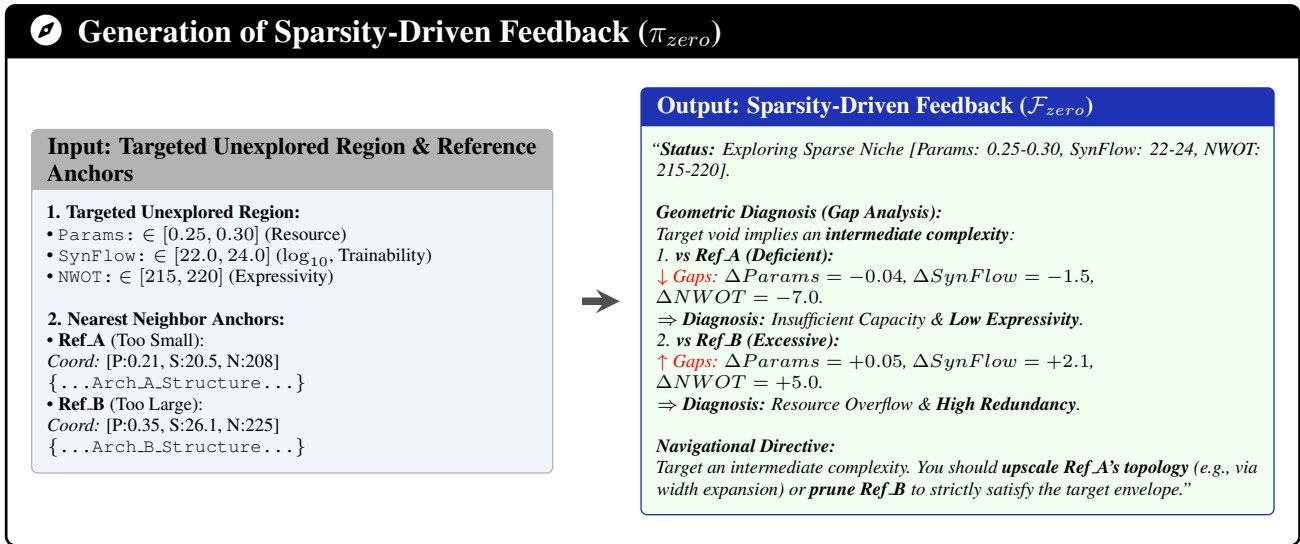

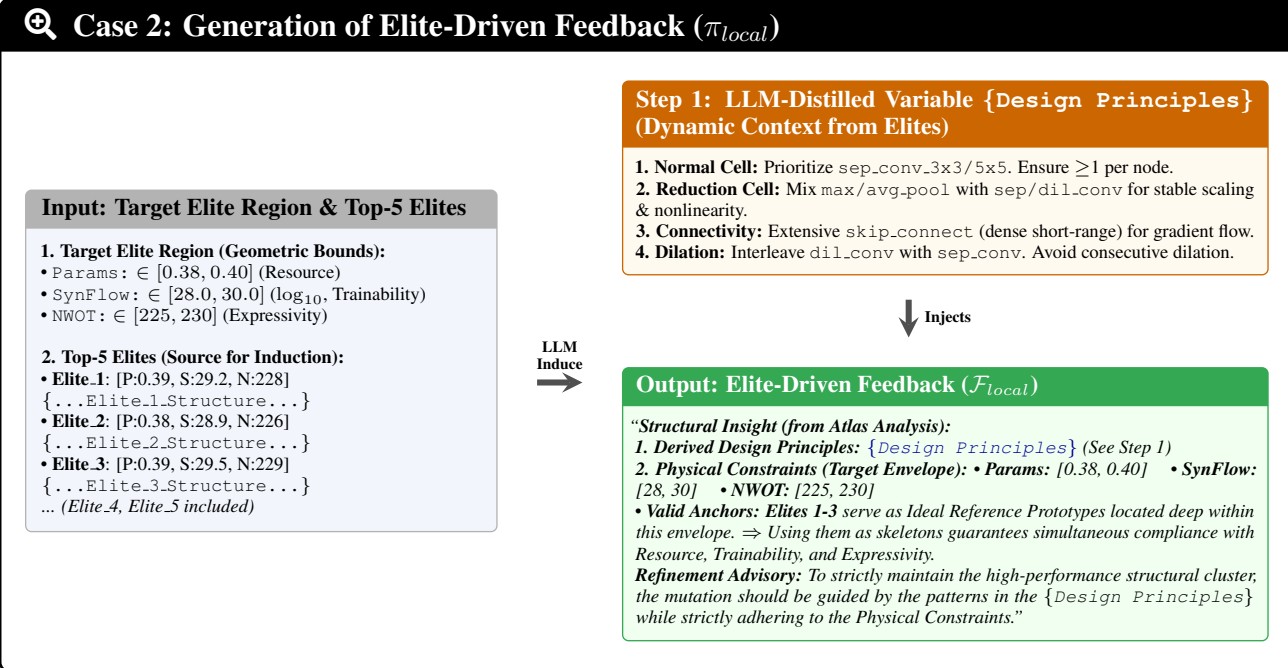

*Figure 14.* Visual Grounding of the Geometric-Semantic Translation Mechanism. (Case 1) Sparsity-Driven Feedback: The translator identifies the *Targeted Unexplored Region* and performs gap analysis against anchors. (Case 2) Elite-Driven Feedback: A Chain-of-Induction process distills abstract *Design Principles* from the *Target Elite Region*, integrating them with physical constraints to guide refinement.

## >_ System Prompt: Design Principle Extraction Protocol

**1. Role & Objective**
**System Role:** Expert Neural Architecture Search (NAS) Architect.
**Objective:** Analyze a provided set of high-performance "Elite Architectures" to distill common structural patterns into actionable *Design Principles*.

**2. Input Context**
The system receives three fundamental context blocks to ground the reasoning:
- **A. Search Space Constraints (Vocabulary):** {operation_constraint}
  *(Constraint: Output principles must ONLY refer to operations defined here.)*
- **B. Topology & Schema Definition (Grammar):** {topology_description}
  *(Dynamic Context: Injects space-specific topological rules (e.g., DARTS vs. NB201 ) to ensure correct structural parsing.)*
- **C. Elite Architectures (Data):** {architectures_text}
  *(Source: A list of Top-K architectures with high validation accuracy.)*

**3. Chain-of-Induction Logic**
The LLM is instructed to perform step-by-step inductive reasoning:
- **Step 1 (Frequency Analysis):** Identify dominant operation types (e.g., kernel sizes, pooling) appearing across elites.
- **Step 2 (Topology Mining):** Analyze node connectivity patterns (e.g., depth, skip-connect density).
- **Step 3 (Abstraction):** Synthesize commonalities into abstract rules.

**4. Required Output Format (Schema)**
Generate a list of 5–10 distinct principles under the header **"Design Principles Summary:"**.

---

**Schema Definition:**
```
N. <Scope (e.g., Topology/Op)>:  <Actionable Rule using Valid Ops>
```

---

**Example:**
"1.  Operation Pattern:  Prioritize sep_conv_3x3 to enhance feature extraction with minimal parameter cost."

*Figure 15.* The Design Principle Extraction Prompt used by the Geometric-Semantic Translator (Case 2). This structural template instructs the LLM to perform inductive reasoning on the Top-5 Elites, distilling their topological commonalities into text-based Design Principles while strictly adhering to the search space definition.

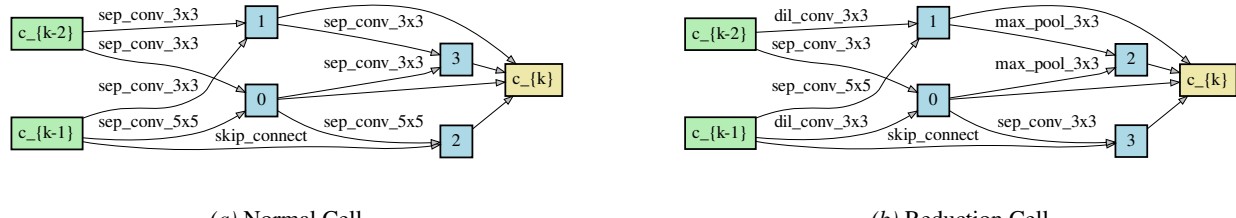

*(a)* Normal Cell                          *(b)* Reduction Cell

*Figure 16.* DARTS cell architecture found by SAGE-NAS on ImageNet dataset with model size 5.88 MB.

