# OpenReview forum: "SAGE-NAS: Synergizing LLM-Based Semantic Agent with Graph-Based Evaluator for Neural Architecture Search"
_ICML.cc/2026/Conference — ICML 2026 regular_

### Official Review · Reviewer_wTXm · 2026-03-06

**Soundness:** 2
**Presentation:** 3
**Significance:** 3
**Originality:** 3
**Overall Recommendation:** 4
**Confidence:** 4

**Summary:**

SAGE-NAS proposes a synergistic evolutionary neural architecture search framework designed to address the "semantic-physical misalignment" problem in large language model-guided architecture search. The framework consists of three core components: the Semantic Agent utilizes LLMs to execute context-guided sub-policies for architecture generation; the Dual-Modality Graph Evaluator injects zero-cost proxies (ZCPs) as physical priors into topological features through a cross-attention mechanism, achieving fast and accurate performance estimation; and the Behavioral Atlas employs sparsity-driven exploration to maintain population diversity. SAGE-NAS constructs a closed-loop optimization mechanism that deeply integrates the high-level semantic planning capabilities of LLMs with the low-level physical sensitivity of ZCPs. Experiments on multiple benchmarks including NAS-Bench-201, NAS-Bench-101, DARTS, and TransNAS-Bench-101 demonstrate that this method surpasses existing zero-cost proxy and LLM-NAS approaches in search efficiency, accuracy, and cross-task generalization capability, achieving a 15.17% test error rate (SOTA) on CIFAR-100 with a search cost of only 0.86 GPU days.

**Compliance With Llm Reviewing Policy:**

Affirmed.

**Ethics Expertise Needed:**

["Legal Compliance (e.g., EU AI Act, GDPR, copyright, terms of use)"]

**Key Questions For Authors:**

We truly appreciate the three-dimensional metric space design adopted in your Behavioral Atlas, which has demonstrated effectiveness in preventing mode collapse throughout your experiments. However, we wonder whether this specific dimensional choice—particularly the selection of SynFlow and NWOT—might exhibit inherent biases toward particular search spaces or task types. For instance, in networks requiring extreme sparsity (such as pruned networks) or tasks particularly sensitive to gradient flow (like long-tail distribution data), is there theoretical justification or ablation studies demonstrating that this specific combination of dimensions (SynFlow, NWOT, and resource metrics) constitutes the optimal or most compact basis for characterizing the architectural behavior space? If alternative zero-cost proxy combinations (such as GraSP + Jacov) were substituted, would the coverage efficiency of the Behavioral Atlas and the final search performance change? Furthermore, do you envision incorporating an adaptive dimension selection mechanism in future work, one that dynamically adjusts the composition of the behavior space based on task feedback, thereby enhancing the framework's generalization capability across diverse scenarios?

**Limitations:**

1. The selection of Behavioral Atlas axes (SynFlow, NWOT, FLOPs/Params) may not be universally applicable to all task types.
2. The self-supervised pre-training of the Dual-Modality Graph Evaluator requires additional computational resources, and the correlation between the design of pre-training tasks (node degree regression, operation reconstruction) and the downstream NAS objectives awaits theoretical validation.
3.The HyperNetwork-driven dynamic adaptation mechanism increases model parameters, potentially leading to overfitting in small-scale search spaces.
4. Although LLM inference costs are optimized through Qwen-plus, API access remains necessary, which is not conducive to fully offline deployment.
5. The framework's applicability in continuous search spaces or weight-sharing NAS paradigms has not been thoroughly explored.

**Strengths And Weaknesses:**

The SAGE-NAS framework demonstrates several notable strengths, including its deep insight in identifying the semantic-physical misalignment as the core bottleneck of LLM-guided NAS, and its systematically designed closed-loop optimization that synergizes three core modules with rigorous mathematical formalizations for mechanisms such as dual-modality fusion and geometric-semantic translation. The work is supported by comprehensive experiments across multiple benchmarks and multi-modal vision tasks, with thorough ablation studies validating each module's contribution, alongside meticulous engineering implementation featuring self-supervised pre-training, compound loss functions, and robust repair protocols, while maintaining cost-effective LLM inference through Qwen-plus. However, the framework also exhibits certain limitations: its high architectural complexity may increase implementation and maintenance difficulty; despite robustness testing, the inherent stochasticity of LLM inference could affect experimental reproducibility; the empirical selection of Behavioral Atlas dimensions lacks theoretical guidance; cross-task generalization shows slight performance degradation on certain tasks like Layout; and the study does not explore scalability in larger, task-specific search spaces such as those for object detection or semantic segmentation.

---

> ### Author Rebuttal · Authors · 2026-03-30
>
> Thanks a lot for your insightful comments and detailed suggestions. We hope our responses address your concerns and enhance your view of our work.
>
> **1. Selection of the Three Behavioral Axes**
>
> These axes are empirically grounded (NAS-Bench-Suite-Zero [1]) to construct a complementary "Expressivity-Trainability-Resource" 3D space.
>
> * **Fidelity & Complementarity:** NAS-Bench-Suite-Zero [1] empirically demonstrates that NWOT (expressivity) and SynFlow (trainability) exhibit both top-tier correlations with ground-truth accuracy and exceptionally strong complementarity.
> * **Resource Constraint:** Params/FLOPs acts as a hard constraint, preventing the algorithm from "cheating" by infinitely stacking layers to inflate ZCP scores.
>
> **New Ablation (NAS-Bench-201):** To empirically validate this, we compared our configuration against your suggested alternative and a low-complementarity baseline.
>
> | **Behavioral Axes** | **Atlas Coverage (%)** | **CIFAR-10 (%)** | **CIFAR-100 (%)** | **ImageNet-16 (%)** |
> | :--- | :---: | :---: | :---: | :---: |
> | SNIP + Jacov + Params/FLOPs | $16.8 \pm 2.1$ | $93.67 \pm 0.39$ | $71.70 \pm 0.45$ | $45.87 \pm 0.38$ |
> | GraSP + Jacov + Params/FLOPs | $19.1 \pm 1.6$ | $94.14 \pm 0.15$ | $73.09 \pm 0.32$ | $46.22 \pm 0.25$ |
> | **Ours (NWOT + SynFlow + Params/FLOPs)** | $\mathbf{20.3 \pm 1.2}$ | $\mathbf{94.37 \pm 0.00}$ | $\mathbf{73.51 \pm 0.00}$ | $\mathbf{47.05 \pm 0.31}$ |
>
> **2. Clarification on Task Generalization & Adaptive Dimension Plan**
>
> SAGE-NAS actually achieves SOTA on the Room Layout task (Table 3). However, your insightful observation regarding cross-task degradation seems to be reflected in the slight performance fluctuation of the Autoencoder task, which we suspect stems from the inherent biases of fixed ZCPs. We highly value your constructive suggestion of an "adaptive dimension selection" mechanism to break this bottleneck. To operationalize your idea, we have devised a detailed blueprint for the Future Work section of our final manuscript: During the initialization phase, SAGE-NAS will probe a minimal set of anchor architectures (e.g., 50) to obtain ground-truth feedback. We will then execute a bi-objective optimization—maximizing the selected ZCPs' Spearman correlation with the true feedback while minimizing their inter-ZCP Pearson correlation. This low-cost "warm-up" will dynamically assemble the optimal behavioral basis for any unknown task.
>
> **3. Computational Overhead & Pre-training Validation**
>
> * **One-Time Overhead:** Pre-training is strictly a one-time cost per search space. Once topological features are learned, the model seamlessly transfers to all downstream tasks at zero additional cost.
> * **Task Validation:** As shown in the Joint Ablation Study below (Section 4), both structural (node degree) and semantic (operation) reconstructions are indispensable for capturing robust topological priors.
>
> **4. Risk of HyperNetwork Overfitting in Small Spaces**
>
> Rather than simply expanding capacity, the HyperNetwork acts as a structural regularizer. In data-scarce loops, traditional "static adapters" overfit by memorizing the limited sample buffer. Conversely, generating weights dynamically conditioned on topology forces the model to learn a generalized meta-mapping, explicitly preventing rote memorization. As evidenced below, this dynamic adaptation significantly outperforms static counterparts.
>
> **Joint Ablation Study (on NAS-Bench-201):**
>
> | **Ablation Variant** | **CIFAR-10 (%)** | **CIFAR-100 (%)** | **ImageNet-16 (%)** |
> | :--- | :---: | :---: | :---: |
> | w/o Pre-training | $94.06 \pm 0.22$ | $73.21 \pm 0.21$ | $45.97 \pm 0.63$ |
> | w/o Node Degree | $94.21 \pm 0.15$ | $73.33 \pm 0.18$ | $46.07 \pm 0.49$ |
> | w/o Op. Reconstruction | $94.24 \pm 0.12$ | $73.30 \pm 0.21$ | $46.34 \pm 0.37$ |
> | Static Adapter | $94.16 \pm 0.08$ | $73.27 \pm 0.22$ | $46.18 \pm 0.55$ |
> | **Ours (Full Model)** | $\mathbf{94.37 \pm 0.00}$ | $\mathbf{73.51 \pm 0.00}$ | $\mathbf{47.05 \pm 0.31}$ |
>
> **5. API Dependency & Fully Offline Deployment**
>
> SAGE-NAS is intrinsically model-agnostic and does not mandate closed-source APIs. While our main experiments utilized Qwen-plus, Table 10 in our paper already evaluates open-source, locally deployable alternatives. Lightweight models like Qwen3-32B and even Qwen3-8B achieve highly competitive search performance, proving that SAGE-NAS fully supports secure, 100% offline closed-loop deployment.
>
> **6. Applicability in Continuous/Weight-Sharing NAS**
>
> SAGE-NAS employs discrete evolution to avoid the inherent pitfalls of continuous, weight-sharing paradigms (e.g., rank inconsistency and co-adaptation). Nevertheless, our highly competitive results on the **DARTS space (Table 2)** empirically validate its strong effectiveness even in spaces traditionally dominated by continuous methods.
>
> *References:*
>
> [1] Krishnakumar, A., et al. "NAS-Bench-Suite-Zero: Accelerating research on zero cost proxies." NeurIPS 2022.

---

> > ### Author Rebuttal · Reviewer_wTXm · 2026-04-07
> >
> > Thanks for your response. It solves my concern. The score will be kept.

---

> > > ### Author Response · Authors · 2026-04-07
> > >
> > > We sincerely thank you for reviewing our rebuttal and confirming that our responses have resolved your concerns. Your constructive suggestions—particularly regarding the justification of the behavioral axes, the vision for an adaptive dimension selection mechanism, and the prompt for deeper ablations on the Graph Evaluator's components—have been highly valuable in strengthening the rigor of our paper. As promised, we will ensure that all the newly added ablation studies and the detailed blueprint for this adaptive mechanism are carefully incorporated into the final manuscript. Thank you again for your time, effort, and positive support for our work.

---

### Official Review · Reviewer_ZZHs · 2026-03-07

**Soundness:** 2
**Presentation:** 2
**Significance:** 3
**Originality:** 3
**Overall Recommendation:** 5
**Confidence:** 4

**Summary:**

This paper proposes a framework named SAGE-NAS, which synergizes Large Language Models with Neural Architecture Search. The framework consists of three core components: an LLM-Based Semantic Agent responsible for generating candidate architectures; a Dual-Modality Graph Evaluator that fuses topological structures with physical priors for rapid performance prediction; and a Behavioral Atlas that guides the search process to escape local optima. Experimental results demonstrate that this method achieves state-of-the-art performance across multiple standard benchmarks.

**Compliance With Llm Reviewing Policy:**

Affirmed.

**Final Justification:**

The authors' rebuttal completely resolves my concerns. After carefully re-reading the paper, all reviewers' comments, and the authors' rebuttal, I believe this work demonstrates strong novelty and makes significant contributions to advancing the LLM-based NAS field. Therefore, I consider it entirely justified to raise my score for this paper to 5.

**Key Questions For Authors:**

1. Human-designed architectures can also be semantically plausible yet physically difficult to train (many designs before ResNet, for example). Why are LLMs more prone to this problem than humans?
2. The paper selects SynFlow, NWOT, and Params/FLOPs as the three axes to construct the behavioral space. Why these three?
3. ZCPs are compressed into a single vector (64-dimensional) serving as the Query, while the topology has $N$ nodes. This means all ZCP information must pass through a bottleneck to re-weight topological nodes. If the proposed method is applied to search for large-scale architectures, $N$ becomes very large. Will this cause information loss?

**Limitations:**

No. The authors do not discuss limitations. The paper is primarily tested on NAS benchmarks in computer vision, without involving architecture search for natural language processing or other domains. The authors should explicitly point this out in the limitations section.

**Strengths And Weaknesses:**

Strengths:
1. The misalignment between semantics and physics is indeed a core issue in current Large Language Models for Neural Architecture Search, and this paper focuses on resolving this problem, making the research highly valuable.
2. Experimental data shows that this method requires only 0.86 GPU days in the DARTS search space, which is much faster than many traditional methods.
3. The Semantic Agent employs four complementary sub-policies—local exploitation, global exploration, crossover, and zero-shot generation—to balance exploitation and exploration, demonstrating innovative overall framework design.
Weaknesses:
1. The paper describes that the translator deterministically maps geometric deviations into semantic descriptors (Appendix D.3). This design presents issues. On one hand, the translator uses fixed rules (e.g., "$\Delta$Params < 0 $\to$ Insufficient Capacity"), which are manually preset rather than learned from data. Different search spaces may require entirely different mapping logic. On the other hand, the information flow is unidirectional, with only geometric-to-semantic mapping and no semantic-to-geometric feedback. If the architecture generated by the Large Language Model is semantically plausible but geometrically deviates from the target, the system cannot automatically correct the prompt.
2. The maximum dispersion assumption in initialization sampling remains untested. The SelectMaxDispersion strategy assumes that the initial population should cover the entire behavioral space. However, in unimodal fitness landscapes, dispersed sampling is suboptimal. If optimal solutions cluster in a specific region, spending the budget on distant regions is wasteful.
3. Appendix C.1 states that after removing the Semantic Agent, "random mutation + random crossover" is used as a substitute. This is not a fair comparison. The core advantage of SAGE-NAS lies in the semantic reasoning capability of Large Language Models, yet the replacement completely lacks semantic capacity and degenerates into the most basic random search. A more reasonable comparison would be to use the same evolutionary framework but replace the Large Language Model with other generative models (such as VAE or GAN), or use fixed heuristic rules to generate architectures, while keeping other components (the Graph Evaluator and Behavioral Atlas) unchanged.

---

> ### Author Rebuttal · Authors · 2026-03-30
>
> We thank you for the insightful comments. We hope our responses address your concerns.
>
> **1. Design of the Geometric-Semantic Translator**
>
> **(1) Heuristic Mapping:** Predefined rules (e.g., $\Delta \text{Params} < 0 \rightarrow$ "Insufficient Capacity") provide *relative* navigational directives based on invariant physical semantics, rather than absolute performance judgments. The actual architectural efficacy is rigorously determined by the Graph Evaluator.
>
> **(2) Unidirectional Flow:** While single-step reverse prompt correction is absent, our macro-level evolutionary closed-loop inherently compensates. If the LLM repeatedly deviates from a targeted empty Region X, the atlas coverage will naturally stagnate ($\Delta\Omega_t = 0$). This explicitly triggers the Adaptive Spatio-Temporal Strategy Scheduling (Eq. 8), imposing a sparsity penalty that forces the system to prioritize global exploration directives in subsequent iterations until the overall coverage successfully expands ($\Delta\Omega_t > 0$). Thus, population-level macro trial-and-error effectively mitigates the reliance on single-step prompt precision.
>
> **2. Maximum Dispersion in Initialization**
>
> *SelectMaxDispersion* mitigates the "Semantic Collapse" inherent to LLM-based NAS. Since pre-trained LLMs tend to generate homogeneous architectures, we enforce maximum dispersion *only* during initialization (Phase 1) to forcibly break this bias. Subsequently, $\pi_{local}$ efficiently shifts the budget to exploit elite regions. The ablation on NAS-Bench-201 below confirms that omitting this causes premature convergence and degrades performance.
>
> | **Method** | **C-10 (%)** | **C-100 (%)** | **IN-16 (%)** |
> | :--- | :---: | :---: | :---: |
> | w/o SelectMaxDispersion | $94.36 \pm 0.00$ | $73.17 \pm 0.08$ | $46.48 \pm 0.00$ |
> | SAGE-NAS (Ours) | $\mathbf{94.37 \pm 0.00}$ | $\mathbf{73.51 \pm 0.00}$ | $\mathbf{47.05 \pm 0.31}$ |
>
> **3. Ablation Baseline for the Semantic Agent**
>
> **(1) Incompatibility of VAEs/GANs:** VAEs/GANs require massive offline pre-training on thousands of graphs to learn validity and operate in continuous latent spaces. SAGE-NAS operates in a data-scarce, online regime. Introducing a VAE/GAN would require a fundamentally different offline pipeline, violating the premise of a fair, online ablation.
>
> **(2) New Ablation on NAS-Bench-201:** Embracing your constructive alternative, we designed a "Heuristic EA Baseline" to replace the LLM. To guide the search while maintaining population diversity, both its mutation and crossover operations sample or inherit operators with probabilities directly proportional to their historical average scores, completely omitting any semantic CoT.
>
> | **Method** | **C-10 (%)** | **C-100 (%)** | **IN-16 (%)** |
> | :--- | :---: | :---: | :---: |
> | w/o Agent (Random Search) | $94.04 \pm 0.31$ | $71.40 \pm 0.56$ | $45.70 \pm 0.24$ |
> | Heuristic EA Baseline (New) | $94.22 \pm 0.18$ | $72.35 \pm 0.29$ | $46.15 \pm 0.35$ |
> | SAGE-NAS (LLM-Driven) | $\mathbf{94.37 \pm 0.00}$ | $\mathbf{73.51 \pm 0.00}$ | $\mathbf{47.05 \pm 0.31}$ |
>
> **4. Semantic-Physical Misalignment in LLMs vs. Humans**
>
> LLMs are more prone to "semantic-physical misalignment" than humans due to two fundamental gaps.
>
> **(1) Modality:** Humans utilize mathematical deduction, whereas LLMs learn from static codebases, capturing syntactic API co-occurrences rather than continuous physical dynamics (e.g., gradient flow).
>
> **(2) Feedback:** Human design is a "diagnose-and-redesign" closed loop, but standard LLMs are open-loop generators. Without real-time physical feedback, they inevitably produce optimization-hostile "topological hallucinations." SAGE-NAS bridges this gap: our Graph Evaluator and Behavioral Atlas establish a state-aware closed loop, equipping the LLM with the physical diagnostic capabilities it inherently lacks.
>
> **5. Selection of the Three Behavioral Axes (SynFlow, NWOT, Params/FLOPs)**
>
> Thank you for this insightful question. Due to the strict character limit, please refer to our response to Section 1 of our reply to **Reviewer wTXm**.
>
> **6. Potential Information Bottleneck in ZCP Cross-Attention**
>
> **(1) Global Nature of ZCPs:** The 6 ZCPs are inherently macroscopic, network-level scalars, not node-level features. A 64-dimensional space provides abundant capacity to encode the combinatorial states of merely 6 scalars.
>
> **(2) Global-to-Local Modulation:** $Q$ does not need to memorize $N$ nodes. The GCN already models the complex topological relations among $N$ nodes when generating Keys ($K$) and Values ($V$). $Q$ simply acts as a top-down global prompt to contextually re-weight these nodes based on the network's overall physical state.
>
> **7. Limitations and Future Work**
>
> We fully agree and will add a "Limitations and Future Work" section in the final manuscript. Due to character limits, please refer to **Section 2 of our reply to Reviewer Ckqt** for a comprehensive discussion on our framework's boundaries and future directions.

---

> > ### Author Rebuttal · Reviewer_ZZHs · 2026-04-04
> >
> > I thank the authors for their comprehensive rebuttal, which thoroughly addresses my concerns. Overall, I find this work to be highly significant and innovative. Therefore, I am willing to increase my score if possible. I hope the authors will consider incorporating the experimental results presented in the rebuttal into the revised version of the paper (if space permitting).

---

> > > ### Author Response · Authors · 2026-04-04
> > >
> > > We sincerely thank you for your constructive feedback and your positive recognition of our work. We are very glad that our responses and the newly added ablation studies have adequately addressed your concerns. Your insightful suggestions have genuinely helped us improve the rigor and comprehensiveness of our paper. As requested, we will ensure that all the experimental results and discussions presented in this rebuttal are carefully incorporated into the revised manuscript, and we deeply appreciate your willingness to raise the score accordingly. Thank you again for your time and effort in reviewing our work.

---

### Official Review · Reviewer_Ckqt · 2026-03-22

**Soundness:** 3
**Presentation:** 4
**Significance:** 3
**Originality:** 3
**Overall Recommendation:** 4
**Confidence:** 3

**Summary:**

This work combines LM-based evaluation of architectures with zero-cost proxies for Neural Architecture Search. The zero-cost proxies and the architecture graph are fed into a LM to obtain a performance score which is used for evolving neural architectures.

**Compliance With Llm Reviewing Policy:**

Affirmed.

**Key Questions For Authors:**

None

**Limitations:**

The paper does not discuss limitations at all. The empricial evaluation is lacking on several fronts and I wonder about where the methodology could break down, what about, for example, hierarchical search spaces and search on elementary operations?

**Strengths And Weaknesses:**

Strengths
+ Improving the evaluation phase of NAS is a good methodological problem to tackle
+ Combining LLM-based NAS with classic methods (in this case zero-cost proxies) is an interesting methodological direction as the two approaches complement each other
+ The paper is well structured, is easy to follow, and includes helpful figures. The literature and how it relates to this work is covered in sufficient detail.
+ The empirical evaluation includes an ablation study on the core components and type of LLMs

Weaknesses
- The empirical evaluation is severely limited in scope. Few, quite outdated search spaces are considered (NAS-Bench-201, NAS-Bench-101, DARTS, TransNAS-Bench-101), all of which are for CNNs and 3/4 are for image-classificaiton on CIFAR-10/100/ImageNet-16. The methodology having been overfit to these benchmarks is extremely likely, in fact, oracle performance is reached in the case of NAS-Bench-201. If this is improved I would raise my score.
- Limitations are not discussed

---

> ### Author Rebuttal · Authors · 2026-03-30
>
> Thanks a lot for your insightful comments and detailed suggestions. We hope our responses address your concerns and enhance your view of our work.
>
> **1. Expanded Search Spaces: Evaluation on AutoFormer Search Space**
>
> To address your concern regarding empirical scope and potential overfitting on early CNN benchmarks, we have significantly expanded our evaluation to the modern **AutoFormer** search space (Vision Transformers) on the full ImageNet-1k dataset. As shown below, we compare SAGE-NAS with both classic ViT baselines and recent state-of-the-art methods:
>
> | **Algorithms** | **Param (M)** | **Top-1 (%)** | **GPU Days** |
> | :--- | :---: | :---: | :---: |
> | DeiT-Ti [1] | 5.7 | 72.2 | - |
> | TNT-Ti [2] | 6.1 | 73.9 | - |
> | ViT-Ti [3] | 5.7 | 74.5 | - |
> | PVT-Tiny [4] | 13.2 | 75.1 | - |
> | ViTAS-C [5] | 5.6 | 74.7 | 32 |
> | AutoFormer-Ti [6] | 5.7 | 74.7 | 24 |
> | TF-TAS-Ti [7] | 5.9 | 75.3 | 0.5 |
> | Auto-Prox [8] | 6.4 | 75.6 | 0.1 |
> | ParZC [9] | 6.1 | 75.5 | 0.05 |
> | **SAGE-NAS (Ours)** | 7.4 | **75.8** | 0.47 |
>
> Our results demonstrate that SAGE-NAS seamlessly adapts to this new space, discovering a highly competitive ViT architecture that achieves **75.8% Top-1 accuracy** on ImageNet-1k. We believe this successful extension to a Transformer-based space and a massive dataset effectively alleviates the overfitting concerns. These comprehensive results will be incorporated into the final manuscript.
>
> **2. Limitations and Future Work**
>
> We sincerely thank you for pointing out the absence of a limitation discussion. We will explicitly add a "Limitations and Future Work" section in the final manuscript to rigorously analyze our framework's boundaries:
>
> **Limitations of the Current Framework:**
> * **Efficiency Trade-offs vs. Simple ZCPs:** While SAGE-NAS discovers superior final architectures, our evaluation paradigm trades off the "plug-and-play" absolute efficiency of simple ZCPs. A standalone ZCP (e.g., SNIP) scores an architecture in milliseconds via a single forward/backward pass. In contrast, our Evaluator—although highly sample-efficient (requiring no massive upfront labeled data like traditional predictors)—still involves a one-time topological pre-training per search space and online few-shot adaptation. Thus, for strictly resource-constrained edge scenarios requiring instant, zero-warmup scoring, standalone ZCPs remain more computationally advantageous.
> * **Boundaries in Search Space Complexity:**
>     * **Hierarchical Spaces:** While SAGE-NAS can independently search micro-cell topologies or macro-level connectivities, extending to true hierarchical spaces (optimizing both simultaneously as nested graphs) requires modeling complex multi-level dependencies. This challenges our current flat GATv2 Evaluator and increases the LLM's context burden.
>     * **Elementary Operations:** Operating on fine-grained elementary ops (instead of encapsulated macro-ops like `sep_conv_3x3`) would exponentially explode the DAG combinatorial space, potentially exceeding the LLM's symbolic generation capacity and destabilizing the statistical significance of ZCPs.
>
> **Potential Future Work:**
> * **Adaptive Behavioral Atlas Dimensions:** Transitioning to a task-adaptive mechanism. By conducting lightweight task-probing (e.g., ~50 anchor architectures), we plan to execute a bi-objective optimization: maximizing the Spearman correlation of the selected ZCP combination with true task performance while minimizing their Pearson correlation.
> * **Enhancing Local SLM Synergy:** While locally deployable Small Language Models (e.g., Qwen-8B) already show competitive results in SAGE-NAS, they still trail massive closed-source APIs. We plan to explore NAS-specific instruction-tuning for SLMs to bridge this gap, enabling fully offline, high-performance search without privacy concerns.
> * **Hierarchical Graph Encoders:** Upgrading to multi-level GNNs to natively capture both micro and macro architectural dependencies simultaneously, thereby preventing context explosion during LLM reasoning.
>
> *References:*
>
> [1] Touvron, Hugo, et al. "Training data-efficient image transformers & distillation through attention." ICML 2021.
>
> [2] Han, Kai, et al. "Transformer in transformer." NeurIPS 2021.
>
> [3] Dosovitskiy, Alexey, et al. "An image is worth 16x16 words: Transformers for image recognition at scale." ICLR 2021.
>
> [4] Wang, Wenhai, et al. "Pyramid vision transformer: A versatile backbone for dense prediction without convolutions." ICCV 2021.
>
> [5] Su, Xiu, et al. "Vitas: Vision transformer architecture search." ECCV 2022.
>
> [6] Chen, Minghao, et al. "Autoformer: Searching transformers for visual recognition." ICCV 2021.
>
> [7] Zhou, Qinqin, et al. "Training-free transformer architecture search." CVPR 2022.
>
> [8] Wei, Zimian, et al. "Auto-prox: Training-free vision transformer architecture search via automatic proxy discovery." AAAI 2024.
>
> [9] Dong, Peijie, et al. "Parzc: Parametric zero-cost proxies for efficient nas." AAAI 2025.

---

> > ### Author Rebuttal · Reviewer_Ckqt · 2026-03-31
> >
> > My suggestions have been adequately incorporated and I raise my score. I have not read the other reviews and rebuttals.

---

> > > ### Author Response · Authors · 2026-04-02
> > >
> > > We sincerely thank you for your constructive feedback and the decision to raise your score. We are glad that our expanded evaluations on the AutoFormer search space and the discussion on limitations have adequately addressed your concerns. Your insightful suggestions have genuinely helped us improve the rigor and comprehensiveness of our paper. As promised, we will carefully incorporate all the newly added experiments and the "Limitations and Future Work" section into the final manuscript. Thank you again for your time and effort in reviewing our work.

---

### Official Review · Reviewer_Lbwj · 2026-03-23

**Soundness:** 3
**Presentation:** 3
**Significance:** 3
**Originality:** 3
**Overall Recommendation:** 4
**Confidence:** 4

**Summary:**

This paper introduces SAGE-NAS, a closed-loop evolutionary neural architecture search (NAS) framework that integrates a large language model (LLM)-based semantic agent with a graph-based evaluator. The framework is designed to address the semantic-physical misalignment problem in LLM-driven NAS, ensuring that generated architectures are not only logically coherent but also practically trainable.

To achieve this, the authors propose a Dual-Modality Graph Evaluator, which enables rapid and reliable performance estimation by combining architectural topology with physical training priors. In addition, a State-Aware Behavioral Atlas is introduced to mitigate LLM inductive bias and encourage more effective exploration through sparsity-driven strategies.

Extensive experiments across multiple standard search spaces and diverse vision tasks demonstrate that SAGE-NAS consistently outperforms existing zero-cost proxy (ZCP) and LLM-based NAS methods. The results highlight its superior search efficiency, predictive accuracy, and strong generalization across tasks.

**Compliance With Llm Reviewing Policy:**

Affirmed.

**Key Questions For Authors:**

- Could the authors compare to more recent ZCP baselines such as MeCO [1] ?
- Could the authors compare their method with differentiable NAS methods such as DARTS [2], GDAS [3] and DrNAS [4]?
- Could the authors evaluate their method on mobilenetv3 [5] and transformers [6,7,8] search spaces?
- Could the authors evaluate their method with open-source LLMs eg: Llama-3.1 [9] family?

[1] Jiang, T., Wang, H. and Bie, R., 2023. Meco: zero-shot nas with one data and single forward pass via minimum eigenvalue of correlation. Advances in Neural Information Processing Systems, 36, pp.61020-61047.

[2] Liu, H., Simonyan, K. and Yang, Y., DARTS: Differentiable Architecture Search. In International Conference on Learning Representations.

[3] Dong, X. and Yang, Y., 2019. Searching for a robust neural architecture in four gpu hours. In Proceedings of the IEEE/CVF conference on computer vision and pattern recognition (pp. 1761-1770).

[4] Chen, X., Wang, R., Cheng, M., Tang, X. and Hsieh, C.J., DrNAS: Dirichlet Neural Architecture Search. In International Conference on Learning Representations.

[5] Cai, H., Gan, C., Wang, T., Zhang, Z. and Han, S., Once-for-All: Train One Network and Specialize it for Efficient Deployment. In International Conference on Learning Representations.

[6] Chen, M., Peng, H., Fu, J. and Ling, H., 2021. Autoformer: Searching transformers for visual recognition. In Proceedings of the IEEE/CVF international conference on computer vision (pp. 12270-12280).

[7] Wang, H., Wu, Z., Liu, Z., Cai, H., Zhu, L., Gan, C. and Han, S., HAT: Hardware-Aware Transformers for Efficient Natural Language Processing.

[8] Sukthanker, R.S., Zela, A., Staffler, B., Klein, A., Purucker, L., Franke, J.K. and Hutter, F., 2024. HW-GPT-bench: hardware-aware architecture benchmark for language models. Advances in Neural Information Processing Systems, 37, pp.60776-60834.

[9] Grattafiori, A., Dubey, A., Jauhri, A., Pandey, A., Kadian, A., Al-Dahle, A., Letman, A., Mathur, A., Schelten, A., Vaughan, A. and Yang, A., 2024. The llama 3 herd of models. arXiv preprint arXiv:2407.21783.

**Limitations:**

- The authors do discuss potential future work but I did not find a clear description of the limitations of the work
- Could the authors delineate the potential disadvantages of the proposed framework in comparison to simpler zero-cost proxies?

**Strengths And Weaknesses:**

**Strengths**
- The paper is thorough, well-written, and highly accessible. The included visualizations effectively enhance clarity and overall presentation.
- The proposed SAGE-NAS framework, particularly the hypernetwork-based structure of the graph evaluator, is novel and compelling.
- Each component of the method is clearly explained, making the approach easy to follow.
- The experimental evaluation is extensive and comprehensive, covering multiple search spaces and tasks.
- Ablation studies are thoughtfully designed, with clear research questions that help isolate the contribution of each component.
- The appendix provides detailed methodological descriptions and additional evaluations, demonstrating the authors’ rigor.

**Weaknesses**
- The set of baselines and search spaces used for comparison could be expanded to provide a more comprehensive evaluation.
- It would be useful to include comparisons of different baselines in terms of computational cost or FLOPs to contextualize efficiency.
- A comparison with all zero-cost proxies from NAS-Bench-Suite-Zero [1] would strengthen the evaluation and provide a more complete benchmark.

[1] Krishnakumar, A., White, C., Zela, A., Tu, R., Safari, M. and Hutter, F., 2022. Nas-bench-suite-zero: Accelerating research on zero cost proxies. Advances in Neural Information Processing Systems, 35, pp.28037-28051.

---

> ### Author Rebuttal · Authors · 2026-03-30
>
> Thanks a lot for your insightful comments and detailed suggestions. We hope our responses address your concerns and enhance your view of our work.
>
> **1. Expanded Search Spaces**
>
> We agree that evaluating on diverse spaces further solidifies SAGE-NAS's generalizability. Due to stringent rebuttal time constraints, we prioritized the Transformer space (AutoFormer) as it represents a fundamental architectural shift from standard CNNs.
>
> Please refer to **Section 1 of our reply to Reviewer Ckqt** for comprehensive results demonstrating SAGE-NAS's seamless adaptation to this new space. Meanwhile, MobileNetV3 experiments are currently underway and will be fully included in the final manuscript. We hope the highly competitive AutoFormer results already address your concerns regarding our framework's broad applicability.
>
>
> **2. Efficiency Comparison (Search Cost & FLOPs)**
>
> As detailed in our original submission, a baseline comparison regarding search costs (0.86 GPU-days) and model size (5.9M Params) is already provided in **Table 2**.
>
> However, we completely agree with your insightful point that Params alone cannot fully reflect real-world inference efficiency, and FLOPs is indeed a more critical metric. We will explicitly add FLOPs comparisons to Table 2 in the final manuscript. For immediate reference, the SOTA architecture discovered by SAGE-NAS requires **681M FLOPs**, demonstrating highly competitive computational efficiency against existing baselines.
>
> **3. Comprehensive Comparison with All ZCPs and MeCO**
>
> Due to strict space limits, **Table 1** in our paper originally included only representative ZCPs (e.g., SNIP, SynFlow, GraSP) and recent SOTA metrics (ZiCo, AZ-NAS, SWAP). However, we fully agree that a complete benchmark significantly strengthens our evaluation. As you constructively suggested, we have expanded our comparison on the NAS-Bench-201 search space to include additional proxies from NAS-Bench-Suite-Zero and the recent MeCO baseline.
>
> As shown in the supplementary table below, SAGE-NAS consistently outperforms all newly added baselines.
>
> | **Method** | **CIFAR-10 (%)** | **CIFAR-100 (%)** | **ImageNet-16 (%)** |
> | :--- | :---: | :---: | :---: |
> | ZenNAS | $89.55 \pm 1.12$ | $64.69 \pm 3.86$ | $37.18 \pm 3.17$ |
> | NWOT | $91.95 \pm 1.29$ | $68.88 \pm 1.60$ | $42.31 \pm 3.43$ |
> | FLOPs | $93.61 \pm 0.09$ | $70.95 \pm 0.38$ | $41.67 \pm 0.65$ |
> | Params | $93.61 \pm 0.08$ | $70.95 \pm 0.37$ | $41.67 \pm 0.64$ |
> | MeCO | $93.64 \pm 0.31$ | $70.86 \pm 0.96$ | $42.59 \pm 1.77$ |
> | **SAGE-NAS (Ours)** | $\mathbf{94.37 \pm 0.00}$ | $\mathbf{73.51 \pm 0.00}$ | $\mathbf{47.05 \pm 0.31}$ |
>
> **4. Comparison with Differentiable NAS (e.g., GDAS, DrNAS)**
>
> Regarding your point on differentiable NAS, **Table 2** in our paper already compares SAGE-NAS with mainstream differentiable methods (DARTS, PC-DARTS, IS-DARTS) on the DARTS space.
>
> However, we fully agree that incorporating GDAS and DrNAS further enriches the evaluation. As shown below (evaluated on the DARTS space; metrics are error rates), SAGE-NAS maintains highly competitive performance.
>
> | **Method** | **C-10 (%)** | **C-100 (%)** | **IN-1K Top-1 (%)** | **IN-1K Top-5 (%)** | **Params (M)** | **FLOPs (M)** | **Cost (GPU-days)** |
> | :--- | :---: | :---: | :---: | :---: | :---: | :---: | :---: |
> | GDAS | $2.82$ | $18.13$ | $26.0$ | $8.5$ | $5.3$ | $581$ | $0.21$ |
> | DrNAS | $2.46 \pm 0.03$ | - | $\mathbf{23.7}$ | $7.1$ | $5.7$ | $-$ | $4.6$ |
> | **SAGE-NAS (Ours)**| $\mathbf{2.42 \pm 0.06}$ | $\mathbf{15.17}$ | $\mathbf{23.7}$ | $\mathbf{6.8}$ | $5.9$ | $681$ | $0.86$ |
>
> **5. Evaluation with Open-Source LLMs**
>
> We respectfully clarify that SAGE-NAS is intrinsically model-agnostic and does not mandate closed-source APIs. **Table 10** in our paper already demonstrates highly competitive performance using locally deployable open-source variants (e.g., Qwen3-32b, Qwen3-8b).
>
> Per your constructive request, we further evaluated the open-source Llama-3.3-70B-Instruct on the NAS-Bench-201 search space. As shown below, it achieves highly competitive performance, proving that SAGE-NAS consistently maintains robustness and efficacy across entirely different LLM families.
>
> | **LLM Backbone** | **CIFAR-10 (%)** | **CIFAR-100 (%)** | **ImageNet-16 (%)** |
> | :--- | :---: | :---: | :---: |
> | Llama-3.3-70B-Instruct| $94.34 \pm 0.02$ | $72.83 \pm 0.11$ | $46.47 \pm 0.10$ |
> | Qwen-plus (Main Backbone) |$\mathbf{94.37 \pm 0.00}$  | $\mathbf{73.51 \pm 0.00}$  |$\mathbf{47.05 \pm 0.31}$  |
>
>
> **6. Limitations and Future Work**
>
> We sincerely thank you for pointing out the absence of a limitation discussion. We fully agree and will explicitly add a "Limitations and Future Work" section in the final manuscript.
>
> Due to strict character limits here, please kindly refer to **Section 2 of our reply to Reviewer Ckqt**, where we provide a comprehensive discussion on our framework's limitations and future work.

---

> > ### Author Rebuttal · Reviewer_Lbwj · 2026-04-05
> >
> > I thank the authors for their detailed response as well as the additional experiments. I would have appreciated more experiments on Mobilenetv3 search space and the GPT space. I maintain my positive score and think that this paper would be interesting for the ICML audience.

---

> > > ### Author Response · Authors · 2026-04-07
> > >
> > > We sincerely thank you for maintaining your positive score and for your encouraging endorsement of our work! Regarding your follow-up questions:
> > >
> > > **(1) MobileNet Space Evaluation:** Following your valuable suggestion, we investigated the MobileNetV3 space. However, we observed that mainstream NAS research predominantly conducts experiments on the **MobileNetV2 search space** (MBConv-based), which provides a wealth of established baselines for fair comparison. Therefore, to ensure a rigorous evaluation against existing methods, we conducted our experiments on this MobileNetV2 space.
> > >
> > > As shown in the table below, remarkably, even with just a single search run constrained by the brief rebuttal window, SAGE-NAS successfully discovered a highly competitive architecture achieving **80.0% Top-1 accuracy**. This result outperforms recent methods (e.g., ZiCo, AZ-NAS), further demonstrating the strong transferability and stability of our framework. We are currently conducting additional experimental runs to ensure statistical robustness, and we will incorporate the comprehensive results into the final manuscript.
> > >
> > > | Method | FLOPs (M) | Top-1 Acc (%) | Cost (GPU-days) |
> > > | :--- | :---: | :---: | :---: |
> > > | DARTS [1] | 574 | 73.3 | 4 |
> > > | ProxylessNAS [2] | 595 | 76.0 | 8.3 |
> > > | OFA [3] | 662 | 78.7 | 50 |
> > > | ZenNAS [4] | 611 | 79.1 | 0.5 |
> > > | ZiCo [5] | 603 | 79.4 | 0.4 |
> > > | AZ-NAS [6] | 615 | 79.9 | 0.6 |
> > > | **SAGE-NAS (Ours)** | 613 | **80.0** | 0.72 |
> > >
> > > **(2) GPT Space:** We deeply appreciate your forward-looking suggestion. Extending NAS to language model architectures is a highly valuable frontier. However, addressing the unique characteristics of these spaces—such as entirely different operator sets, scaling behaviors, and evaluation metrics—requires extensive framework adaptations and massive computational resources, which makes it practically unfeasible to accomplish within the brief discussion timeframe. We will explicitly add a dedicated discussion in our "Limitations and Future Work" section regarding the potential of extending our framework to NLP architecture search.
> > >
> > > Thank you once again for your valuable suggestions, which have significantly strengthened our evaluation, and for your continued support of our work.
> > >
> > > References:
> > >
> > > [1] Liu, H., et al. "DARTS: Differentiable architecture search." ICLR 2019.
> > >
> > > [2] Cai, H., et al. "ProxylessNAS: Direct neural architecture search on target task and hardware." ICLR 2019.
> > >
> > > [3] Cai, H., et al. "Once-for-all: Train one network and specialize it for efficient deployment." ICLR 2020.
> > >
> > > [4] Lin, M., et al. "Zen-NAS: A zero-shot NAS for high-performance image recognition." ICCV 2021.
> > >
> > > [5] Li, G., et al. "ZiCo: Zero-shot NAS via inverse coefficient of variation on gradients." ICLR 2023.
> > >
> > > [6] Lee, J., et al. "AZ-NAS: Assembling zero-cost proxies for network architecture search." CVPR 2024.

---

### Decision · Program_Chairs · 2026-04-30

[review text omitted: it was posted to a different submission]